# Dynamically-enhanced strain in atomically thin resonators

Xin Zhang [1✉], Kevin Makles[1], Léo Colombier[1], Dominik Metten[1], Hicham Majjad[1], Pierre Verlot [2,3] &
Stéphane Berciaud [1,3✉]

Graphene and related two-dimensional (2D) materials associate remarkable mechanical, electronic, optical and phononic properties. As such, 2D materials are promising for hybrid systems that couple their elementary excitations (excitons, phonons) to their macroscopic mechanical modes. These built-in systems may yield enhanced strain-mediated coupling compared to bulkier architectures, e.g., comprising a single quantum emitter coupled to a nano-mechanical resonator. Here, using micro-Raman spectroscopy on pristine monolayer graphene drums, we demonstrate that the macroscopic flexural vibrations of graphene induce dynamical optical phonon softening. This softening is an unambiguous fingerprint of dynamically-induced tensile strain that reaches values up to $\approx 4 \times 10^{-4}$ under strong non-linear driving. Such non-linearly enhanced strain exceeds the values predicted for harmonic vibrations with the same root mean square (RMS) amplitude by more than one order of magnitude. Our work holds promise for dynamical strain engineering and dynamical strain-mediated control of light-matter interactions in 2D materials and related heterostructures.

[1] Université de Strasbourg, CNRS, Institut de Physique et Chimie des Matériaux de Strasbourg, UMR 7504, F-67000 Strasbourg, France. [2] School of Physics and Astronomy, University of Nottingham, Nottingham NG7 2RD, United Kingdom. [3] Institut Universitaire de France, 1 rue Descartes, 05 75231 Paris Cedex, France. ✉email: zhxsemi@gmail.com; stephane.berciaud@ipcms.unistra.fr

Since the first demonstration of mechanical resonators made from suspended graphene layers[1], considerable progress has been made to conceive nano-mechanical systems based on 2D materials[2,3] with well-characterised performances[4–8], for applications in mass and force sensing[9] but also for studies of heat transport[10,11], non-linear mode coupling[12–14] and opto-mechanical interactions[5,15,16]. These efforts triggered the study of 2D resonators beyond graphene, made for instance from transition metal dichalcogenide layers[8,11,17,18] and van der Waals heterostructures[19–21]. In suspended atomically thin membranes, a moderate out-of-plane stress gives rise to large and swiftly tunable strains, in excess of 1%[22,23], opening numerous possibilities for strain-engineering[24]. These assets also position 2D materials as promising systems to achieve enhanced strain-mediated coupling[25–28] of macroscopic flexural vibrations to quasiparticles (excitons, phonons) and/or degrees of freedom (spin, valley). Such developments require sensitive probes of dynamical strain. Among the approaches employed to characterise strain in 2D materials, micro-Raman scattering spectroscopy[29] stands out as a local, contactless and minimally invasive technique that has been extensively exploited in the static regime to quantitatively convert the frequency softening or hardening of the Raman active modes into an amount of tensile or compressive strain, respectively[30–33]. Recently, the interplay between electrostatically-induced strain and doping has been probed in the static regime in suspended graphene monolayers[34]. Dynamically-induced strain has been investigated using Raman spectroscopy in bulkier micro electro-mechanical systems[35,36], including mesoscopic graphite cantilevers[37] but remains unexplored in resonators made from 2D materials.

In this article, using micro-Raman scattering spectroscopy in resonators made from pristine suspended graphene monolayers, we demonstrate efficient strain-mediated coupling between "built-in" quantum degrees of freedom (here the Raman-active optical phonons of graphene) of the 2D resonator, and its macroscopic flexural vibrations. The dynamically-induced strain is quantitatively determined from the frequency of the Raman-active modes and is found to attain anomalously large values, exceeding the levels of strain expected under harmonic vibrations by more than one order of magnitude. Our work introduces resonators made from graphene and related 2D materials as promising systems for hybrid opto-electro-mechanics[38] and dynamical strain-mediated control of light-matter interactions.

## Results

**Measurement scheme.** As illustrated in Fig. 1a, the system we have developed for probing dynamical strain in the 2D limit is a graphene monolayer, mechanically exfoliated and transferred as is onto a pre-patterned Si/SiO$_2$ substrate. The resulting graphene drum is capacitively driven using a time-dependent gate bias $V_g(t) = V_{dc} + V_{ac} \cos \Omega t$, with $V_{ac} \ll V_{dc}$ and $\Omega/2\pi$ the drive frequency. The DC component of the resulting force ($\propto V_g^2$, see "Methods" section) enables to control the electrostatic pressure applied to the graphene membrane (and hence its static deflection $\xi$, see Fig. 1a), whereas the AC bias leads to a harmonic driving force ($\propto V_{dc} V_{ac} \cos \Omega t$). A single laser beam is used to interferometrically measure the frequency-dependent mechanical susceptibility at the drive frequency, akin to ref. [1] and, at the same time, to record the micro-Raman scattering response of the atomically thin membrane. We have chosen electrostatic rather than photothermal actuation[39] to attain large RMS amplitudes while at the same time avoiding heating and photothermal backaction effects[10,11], possibly leading to additional damping[8], self-oscillations[10], mechanical instabilities and sample damage. All measurements were performed at room temperature under

high vacuum (see "Methods" section and Supplementary Notes 1 to 8).

**Raman spectroscopy in strained graphene.** The Raman spectrum of graphene displays two main features: the G mode and the 2D mode, arising from one zone-centre (that is, zero momentum) phonon and from a pair of near-zone edge phonons with opposite momenta, respectively (see Fig. 1a and Supplementary Note 1)[29]. Both features are uniquely sensitive to external perturbations. Quantitative methods have been developed to unambiguously separate the share of strain, doping, and possibly heating effects that affect the frequency, full width at half maximum (FWHM) and integrated intensity of a Raman feature[31,34,40–42] (hereafter denoted $\omega_i$, $\Gamma_i$, $I_i$, respectively, here with $i =$ G, 2D). Biaxial strain is expected around the centre of circular graphene drums[22] and the large Grüneisen parameters of graphene ($\gamma_G = 1.8$ and $\gamma_{2D} = 2.4$, with $\gamma_i = \frac{1}{2\omega_i} \frac{\partial \omega_i}{\partial \varepsilon}$ and $\varepsilon$ the level of biaxial strain)[31,32] allow detection of strain levels down to a few $10^{-5}$. The characteristic slope $\frac{\partial \omega_{2D}}{\partial \omega_G} \approx 2.2$ in graphene under biaxial strain is much larger than in the case of electron or hole doping, where the corresponding slope is significantly smaller than 1[41,42]. This difference allows a clear disambiguation between strain and doping (see "Methods" section for details).

**Mechanical and Raman characterisation.** Fig. 1b presents the main characteristics of a circular graphene drum (device 1) in the linear response regime. A Lorentzian mechanical resonance is observed at $\Omega_0/2\pi \approx 33.8$ MHz for $V_{dc} = -6$ V (Fig. 1b and Supplementary Notes 5 and 6). The mechanical mode profile shows radial symmetry (inset in Fig. 1b) as expected for the fundamental flexural resonance of a circular drum[6]. The mechanical resonance frequency is widely gate-tunable: it increases by ~70% as $|V_{dc}|$ is ramped up to 10 V and displays a symmetric, "U-shaped" behaviour with respect to a near-zero DC bias $V_{dc}^0 = 0.75$ V, at which graphene only undergoes a built-in tension. These two features are characteristic of a low built-in tension[4,8,10,43] that we estimate to be $T_0 = (4 \pm 0.4) \times 10^{-2}$ Nm$^{-1}$, corresponding to a built-in static strain $\varepsilon_s^0 = T_0(1 - \nu)/E_{1LG} \approx (1.0 \pm 0.1) \times 10^{-4}$, where $E_{1LG} = 340$ Nm$^{-1}$, $\nu = 0.16$ are the Young modulus and Poisson ratio of pristine monolayer graphene[44] (Supplementary Note 6). The quality factor $Q$ is high, in excess of 1500 near charge neutrality. As $|V_{dc}|$ increases, $Q$ drops down to ~200 due to electrostatic damping[8].

Fig. 1d shows that the Raman response of suspended graphene is tunable by application of a DC gate bias, as extensively discussed in ref. [34]. Once $V_{dc}$ is large enough to overcome $\varepsilon_s^0$, the membrane starts to bend downwards and the downshifts of the G-mode and 2D-mode features measured at the centre of the drum are chiefly due to biaxial strain ($\partial \omega_{2D}/\partial \omega_G \approx 2.2$, see inset in Fig. 1d) with negligible contribution from electrostatic doping[34] (see "Methods" section for details). At $V_{dc} = -9$ V, the $4 \pm 0.5$ cm$^{-1}$ 2D-mode downshift relative to its value near $V_{dc}^0$ yields a gate-induced static strain $\varepsilon_s = 3 \pm 0.3 \times 10^{-4}$ that agrees qualitatively well with the value $\varepsilon_s = 2 \pm 0.2 \times 10^{-4}$ estimated from the gate-induced upshift of $\Omega_0$ (Fig. 1c and Supplementary Note 6). This agreement justifies our assumption that the Young's modulus of our drum is close to that of pristine graphene (see also Supplementary Note 5 for details on the drum effective mass).

Noteworthy, optical interference effects cause a large gate-dependent modulation of $I_G$ and $I_{2D}$ (refs. [31,34] and see normalisation factors in Fig. 1d). Both strain-induced Raman shifts and Raman scattering intensity changes are exploited to

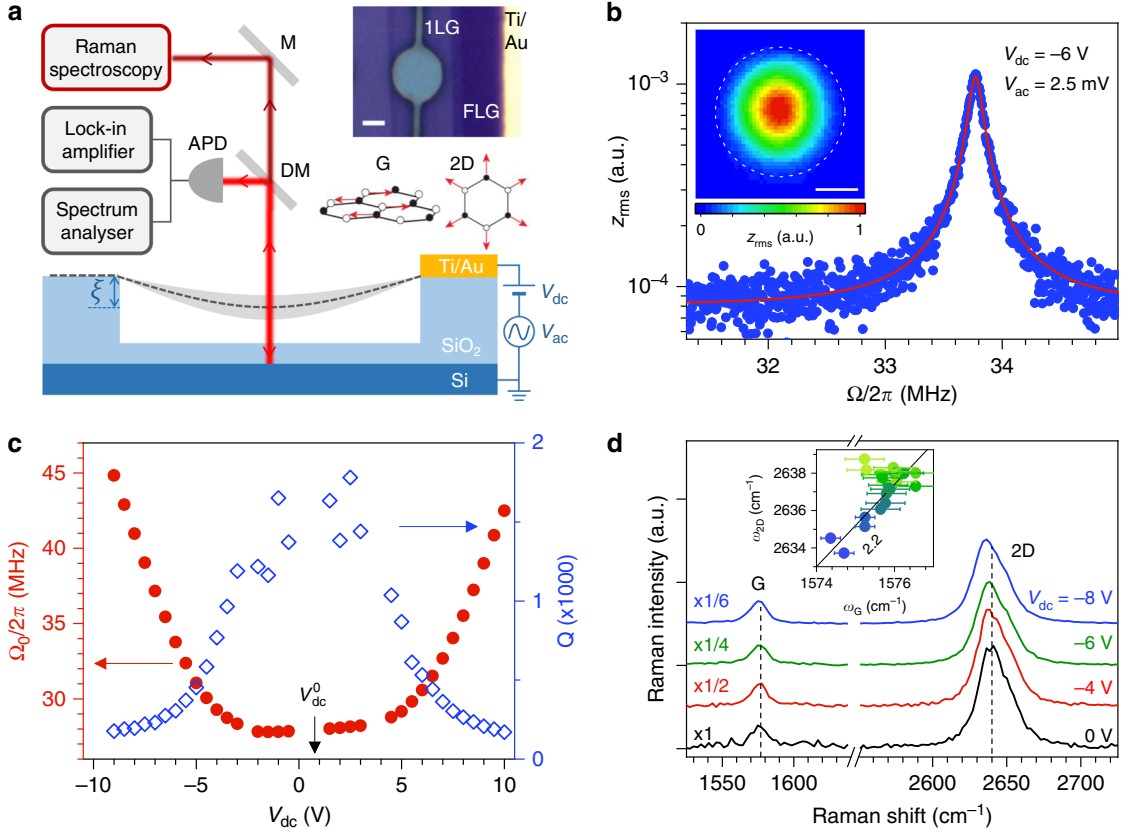

**Fig. 1 Experimental setup and characterisation of pristine graphene drums. a** Sketch of our experiment combining electrostatic actuation, optical readout of the displacement and micro-Raman spectroscopy of a circular graphene drum (device 1). The graphene layer (with its static displacement $\xi$) is represented by the dark grey dashed line; its flexural motion is sketched with the light grey shade. M, DM, APD represent a mirror, a dichroic mirror, an avalanche photodiode, respectively. Upper inset: optical image of a suspended graphene monolayer (1L) contacted by a Ti/Au lead (scale bar: 2 μm). A thicker, few-layer graphene flake (FLG) is also visible. Lower inset: sketch of the atomic displacements contributing to the Raman G mode and 2D mode. **b** RMS mechanical amplitude $z_{rms}$ (blue dots) as a function of the drive frequency $\Omega/2\pi$ at $V_{dc} = -6$ V and $V_{ac} = 2.5$ mV. The red line is a fit based on linear response theory (Supplementary Note 6). Inset: map of the resonant mechanical (scale bar: 2 μm). **c** Resonance frequency $\Omega_0/2\pi$ and corresponding quality factor Q as a function of $V_{dc}$, with $V_{dc}^0$ indicating the charge neutrality point in graphene. **d** Raman spectra measured at the centre of the drum at $V_{dc}$ = 0, −4, −6, −8 V and $V_{ac}$ = 0 mV. Inset: correlation between the G-mode and 2D-mode frequencies ($\omega_{2D}$ and $\omega_G$), extracted from Raman spectra measured with $V_{dc}$ varying from −9 V to 10 V. The light green-to-blue colour scale in circles encodes the increase of $|V_{dc} - V_{dc}^0|$. The straight black line with a slope of 2.2 is a guide to the eye corresponding to strain-induced phonon softening.

consistently estimate that $\xi$ increases from about 30 nm to 70 nm when $V_{dc}$ is varied from −5 V to −9 V (Supplementary Notes 2, 3 and 4).

**Non-linear mechanical response**. We are now examining how the dynamically-induced strain can be readout by means of Raman spectroscopy. First, to obtain a larger sensitivity towards static strain (Supplementary Note 3), we apply a sufficiently high $V_{dc}$ to reach a sizeable $\xi$. $V_{ac}$ is then ramped up to yield large RMS amplitudes. After calibration of our setup (Supplementary Note 5), we estimate that resonant RMS amplitudes $z_{rms}^0$ up to ~10 nm are attained in device 1 (Figs. 2 and 3). In this regime, graphene is a strongly non-linear mechanical system that can be described to lowest order by a Duffing-like equation[5,7,45]:

$$\ddot{z} + \frac{\Omega_0}{Q}\dot{z} + \Omega_0^2 z + \widetilde{\alpha}_3 z^3 = \frac{\widetilde{F}_{el}}{\widetilde{m}}\cos(\Omega t), \quad (1)$$

where $z$ is the mechanical displacement at the membrane centre relative to the equilibrium position $\xi$, $\Omega_0/2\pi$ is the resonance frequency in the linear regime, Q is the quality factor and $\Omega_0/Q$ is the linear damping rate. The effective mass $\widetilde{m}$ and effective applied electrostatic force $\widetilde{F}_{el}$ account for the mode profile of the

fundamental resonance in a rigidly clamped circular drum[6,46] (see "Methods" section and Supplementary Note 6). The linear spring constant is $\widetilde{m}\,\Omega_0^2$. Mechanical non-linearities are considered using an effective third-order term $\widetilde{\alpha}_3$ that changes sign at large enough $\xi$, leading to a transition from non-linear hardening to non-linear softening[5]. Such a behaviour is indeed revealed in our experiments, as shown in Figs. 2a and 3a, where non-linear softening and non-linear hardening are observed at $V_{dc} = -8$ V and $V_{dc} = -6$ V, respectively. At $V_{dc} = -7$ V, we observe a $V_{ac}$-dependent softening-to-hardening transition (Supplementary Notes 6 and 7).

**Dynamical optical phonon softening**. Fig. 2c–e shows the frequencies, linewidths and integrated intensities of the Raman features measured at $V_{dc} = -8$ V (where $\xi \approx 60$ nm), with $V_{ac}$ increasing from 0 to 150 mV and applied at a drive frequency that tracks the $V_{ac}$-dependent non-linear softening of the mechanical resonance frequency $\widetilde{\Omega}_0/2\pi$, that is the so-called backbone curve in Fig. 2a, f (Supplementary Note 6). Both G-mode and 2D-mode features downshift as the drum is non-linearly driven. This phonon softening is accompanied by spectral broadening by up to ~10−15% (Fig. 2d) that increases

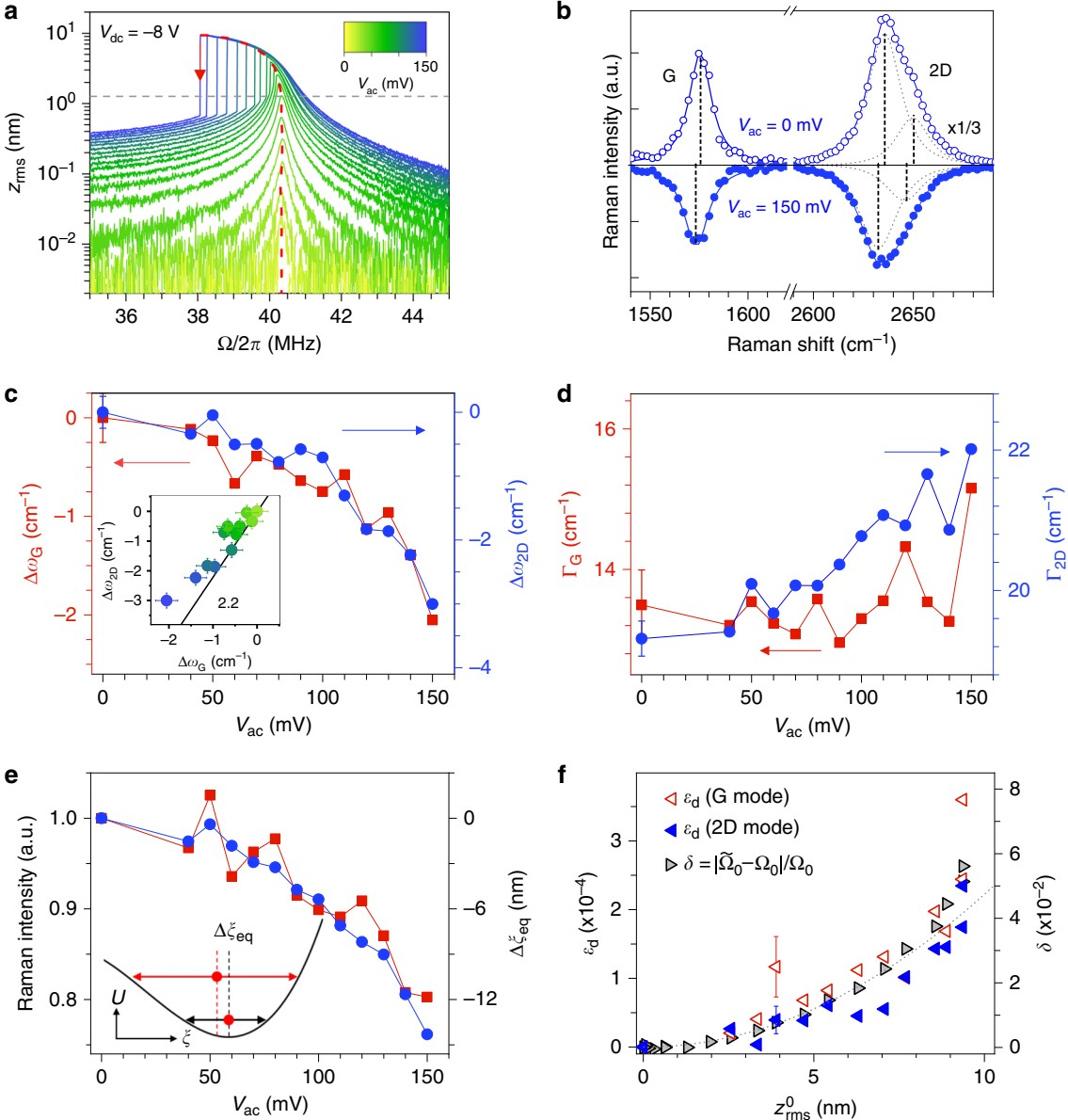

**Fig. 2 Evidence for dynamically-enhanced strain in graphene.** All measurements are performed on device 1 at $V_{dc} = -8$ V. **a** Calibrated RMS mechanical amplitude at the drive frequency $\Omega/2\pi$ ($z_{rms}$) recorded as the frequency is swept downwards, for $V_{ac}$ increasing from 0 to 150 mV. The red dashed line is the backbone curve evidencing non-linear resonance frequency softening. The red arrow indicates the jump-down frequency at $V_{ac} = 150$ mV. The grey dashed line denotes the onset of non-linearity. **b** Raman spectra measured under $V_{ac} = 0$ mV (open symbols and fit) and 150 mV (filled symbols and fit, vertically flipped for clarity). **c, d** G-mode and 2D-mode frequency shifts $\Delta\omega_{G,2D}$ and full-width at half maximum variations ($\Delta\Gamma_{G,2D}$), relative to the values at $V_{ac} = 0$ mV, as a function of $V_{ac}$. Inset in **c**: correlation between $\Delta\omega_{2D}$ and $\Delta\omega_G$. The symbol colour encodes the increase of $V_{ac}$ as in **a**. The straight black line with a slope of 2.2 is a guide to the eye for strain-induced phonon softening. **e** Normalised integrated intensity of G-mode and 2D-mode features as a function of $V_{ac}$. The inset illustrates the equilibrium position shift ($\Delta\xi_{eq}$ between the two red circles) in the non-linear regime, with $U(\xi)$ the potential energy. **f** Time-averaged dynamical strain $\varepsilon_d$ extracted from the softening of G-mode and 2D-mode features (open red and filled blue triangles, respectively) as a function of the corresponding $z_{rms}^0$. The right axis (grey triangles) shows the relative non-linear mechanical resonance frequency shift $\delta = |\widetilde{\Omega}_0 - \Omega_0|/\Omega_0$, where $\widetilde{\Omega}_0$ is the jump-down frequency in **a**. The grey dashed line is a parabolic fit (Supplementary Note 6). Error bars in **c, d, f** are extracted from the fits of Raman spectra. Only one error bar is included in each plot for visibility.

with $z_{rms}$. The correlation plot between $\omega_{2D}$ and $\omega_G$ reveals a linear slope near 2 (see also Supplementary Note 1), which is a characteristic signature of tensile strain[31,41] that gets as high as $\approx 2.5 \times 10^{-4}$ for $z_{rms} \approx 9$ nm.

In Fig. 3a, we compare, on device 1, the frequency-dependence of $z_{rms}$ to that of $\omega_{G,2D}$ and $I_{2D}$, for upward and downward sweeps under $V_{dc} = -6$ V and $V_{ac} = 100$ mV. As in Fig. 2c, sizeable G-mode and 2D-mode softenings are observed near the mechanical resonance (Fig. 3a–c) and assigned to tensile strain

(see correlation plot in Fig. 3c). Remarkably, the hysteretic behaviour of the mechanical susceptibility, associated with non-linear hardening at $V_{dc} = -6$ V, is well-imprinted onto the frequency-dependence of $\omega_{G,2D}$ and $I_{2D}$. Looking further at Fig. 3a, we notice that while $z_{rms}$ fully saturates at drive frequencies above 33.5 MHz and ultimately starts to decrease near the jump-down frequency, the tensile strain keeps increasing linearly up to $\approx 2.5 \times 10^{-4}$ as $\Omega/2\pi$ is raised from 33.2 MHz up to 34.5 MHz.

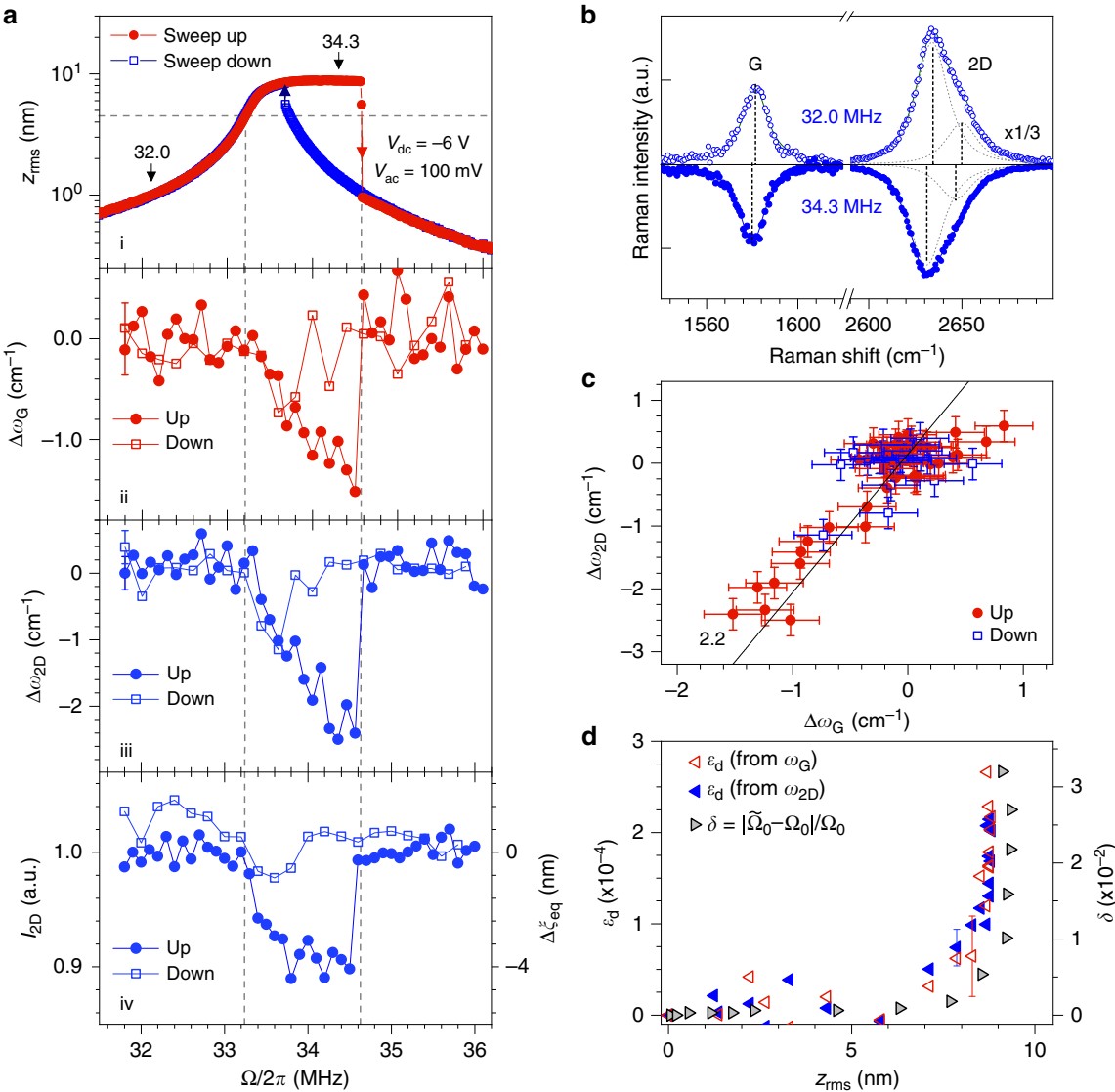

**Fig. 3 Frequency-dependent dynamically-induced strain.** All measurements are performed on device 1 under $V_{dc} = -6\,V$ **a** Panel i: RMS mechanical amplitude $z_{rms}$ as a function of the drive frequency $\Omega/2\pi$ under $V_{ac} = 100\,mV$. Red filled circles and blue open squares indicate upward and downward frequency sweeps. The blue and red arrows denote the jump-down and jump-up frequencies in the non-linear hardening region, respectively. The dashed lines are guides to the eye. Panels ii and iii: Raman frequency shifts $\Delta\omega_G$ and $\Delta\omega_{2D}$ as a function of $\Omega/2\pi$. Panel iv: integrated intensity of the 2D-mode feature as a function of $\Omega/2\pi$. Filled and open symbols in panels 2-4 correspond to upward and downward frequency sweeps, respectively. **b** Raman spectra recorded at $\Omega/2\pi = 32\,MHz$ (open symbols and fit) and 34.3 MHz (filled symbols and fit, vertically flipped for clarity), see arrows in **a**. **c** Correlation between the frequency shifts $\Delta\omega_G$ and $\Delta\omega_{2D}$, relative to the values recorded away from the mechanical resonance. The straight black line with a slope of 2.2 is a guide to the eye for the strain-induced phonon softening. **d** Time-averaged dynamical strain $\varepsilon_d$ extracted from the softening of G-mode and 2D-mode features in **a**-ii and **a**-iii (open red and filled blue triangles, respectively) as a function of the RMS amplitude $z_{rms}$. The right axis (grey triangles) shows the relative non-linear mechanical resonance frequency shift $\delta = \left|\widetilde{\Omega}_0 - \Omega_0\right|/\Omega_0$, where $\widetilde{\Omega}_0$ is the jump-down frequency (see **a** and Supplementary Note 6). Error bars in **a**, **c**, **d** are extracted from the fits of Raman spectra. Only one error bar is included in **a** and **d** for visibility.

**Equilibrium position shift**. As our graphene drums are non-linearly driven, including beyond the Duffing regime (Fig. 3a and Supplementary Notes 6 and 7), the large strains revealed in Figs. 2 and 3 could in part arise from an equilibrium position shift $\Delta\xi_{eq}$ due to symmetry breaking non-linearities[45,47] (inset in Fig. 2e). This effect can be quantitatively assessed through analysis of $I_{G,2D}$. As shown in Fig. 2e both $I_{2D}$ and $I_G$ decrease by about ~20% as $V_{ac}$ increases up to 150 mV. These variations are assigned to optical interference effects (refs. [31,34]); in our experimental geometry they indicate an equilibrium position upshift $\Delta\xi_{eq}$ by up to ≈12 nm (Fig. 2e and Supplementary Note 4), that leads to a reduction of the static tensile strain $\Delta\varepsilon_s \approx 1 \times 10^{-4}$, in stark contrast with the enhanced tensile strain unambiguously

revealed in Fig. 2c. Similarly, the ≈10% drop in $I_{2D}$ near the jump-down frequency at 34.5 MHz indicates an equilibrium position upshift $\Delta\xi_{eq} \approx 4\,nm$ that is qualitatively similar to the results in Fig. 2e. The larger $\Delta\xi_{eq}$ measured at $V_{dc} = -8\,V$ is consistent with our observation of non-linear mechanical resonance softening (Fig. 2a) due to an increased contribution from symmetry breaking non-linearities at large $\xi$ (refs. [5,45,47] and Supplementary Note 6). From these measurements, we conclude that the dynamical softening of $\omega_G$ and $\omega_{2D}$ is not due to an equilibrium position shift.

**Evidence for dynamically-induced strain**. We therefore conclude that the tensile strain measured in device 1 is dynamically-

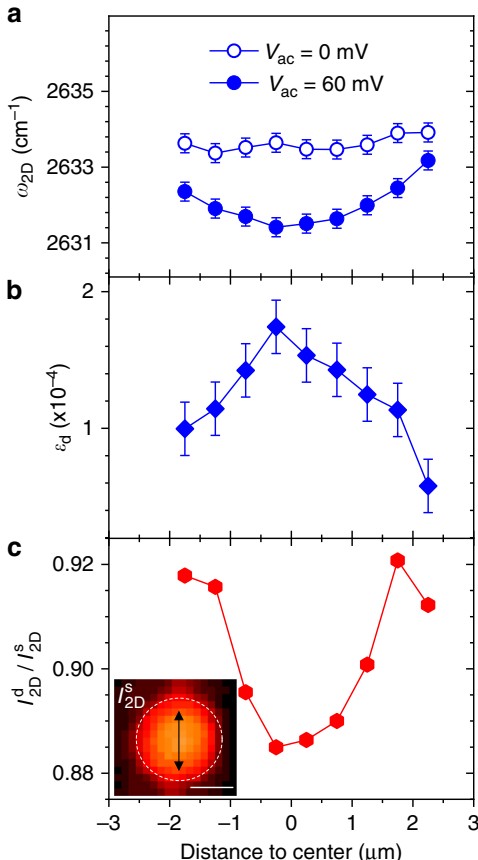

**Fig. 4 Mapping dynamically-induced strain. a** Frequency of the Raman 2D mode along the cross-sections highlighted in **c** in a graphene drum (device 2, 3 μm radius) at $V_{dc} = -6$ V and $V_{ac} = 0$ mV (open symbols) and $V_{ac} = 60$ mV (full symbols). **b** Dynamical strain $\varepsilon_d$ obtained from the difference of the data in **a**. **c** Ratio of the Raman 2D-mode intensity in the driven ($I_{2D}^d$) and static ($I_{2D}^s$) cases. Inset: Map of the Raman 2D-mode intensity $I_{2D}^s$ recorded on the graphene drum (see white dashed contour), at $V_{dc} = -6$ V and $V_{ac} = 0$ V. The double arrow indicates the location of the line scan. The scale bar is 3 μm.

induced (hereafter denoted $\varepsilon_d$) and arises from the time-averaged resonant vibrations of the graphene drum. Starting from a reference recorded at $V_{dc} = -8$ V and $V_{ac} = 0$ mV, $\varepsilon_d$ recorded under resonant driving at $V_{ac} = 150$ mV (where $z_{rms} \approx 9$ nm) is as high as the static strain $\varepsilon_s$ induced when ramping $V_{dc}$ from 0 V to $-8$ V (where $\xi \approx 60$ nm). Along these lines, the small yet observable broadenings $\Delta\Gamma_{G,2D}$ of the Raman features (Fig. 2d) can be assigned to time-averaged Raman frequency shifts due to dynamical strain[48]. We have consistently observed dynamically-enhanced strain in three graphene drums with similar designs, denoted device 1, 2, 3. Complementary results are reported in Supplementary Note 9 for device 1 and in Supplementary Notes 10 and 11 for devices 2 and 3, respectively. In device 3, we have measured $\varepsilon_d \approx 4 \times 10^{-4}$ for $z_{rms} \approx 14$ nm.

**Spatially-resolved dynamically-induced strain.** Interestingly, our diffraction-limited Raman readout enables local mapping of $\varepsilon_d$. Fig. 4 compares $\omega_{2D}$ and $I_{2D}$ recorded across the diameter of a graphene drum (device 2, similar to device 1) under $V_{dc} = -6$ V with and without resonant driving. Very similar results are observed when performing a line-scan along the perpendicular direction (Supplementary Note 9). In the undriven case, we find a nearly flat $\omega_{2D}$ profile, which is consistent with the difficulty in resolving low-levels of static strain below $1 \times 10^{-4}$. In contrast,

finite $\varepsilon_d$ (Fig. 4b) and equilibrium position upshift (Fig. 4c) are observed at the centre of the drum, as in Figs. 2 and 3. We find that $\varepsilon_d$ and the equilibrium position upshift decrease as they are probed away from the centre of the drum and the spatial profile of $\varepsilon_d$ resembles the static tensile strain profile measured on bulged graphene blisters, where strain is biaxial at the centre of the drum and radial at the edges[49].

**Dynamically-enhanced strain.** It is instructive to compare the measured $\varepsilon_d$ to $\varepsilon_d^h = 2/3(z_{rms}/a)^2$, with $a$ the drum radius, the time-averaged dynamically-induced strain estimated for an harmonic oscillation with RMS amplitude $z_{rms}$ (Supplementary Note 7). For the largest $z_{rms} \approx 9$ nm attained in device 1, $\varepsilon_d^h \approx 6 \times 10^{-6}$, i.e., about 40 times smaller than the measured $\varepsilon_d$ (Figs. 2f and 3d). Under strong non-linear driving, we expect sizeable Fourier components of the mechanical amplitude at harmonics of the drive frequency, which could in part be responsible for the large discrepancy between $\varepsilon_d$ and $\varepsilon_d^h$. Harmonics are indeed observed experimentally in the displacement power spectrum of our drums (Supplementary Note 10, device 2) but display amplitudes significantly smaller than the linear component at the drive frequency. In addition, we do not observe any measurable fingerprint of internal resonances[12–14] in the displacement power spectrum.

To get further insights into the unexpectedly large $\varepsilon_d$ deduced from the G-mode and 2D-mode downshifts we plot $\varepsilon_d$ as a function of the corresponding $z_{rms}$ at the centre of the drum (Figs. 2f and 3d). This plot is directly compared to the backbone curves that connect the resonant $z_{rms}$ to the non-linear relative resonance frequency shift $\delta = |\widetilde{\Omega}_0 - \Omega_0|/\Omega_0$, where $\widetilde{\Omega}_0$ is considered equal to the measured jump-down frequency (Figs. 2a, 3a and Supplementary Notes 6 and 7). Remarkably, $\varepsilon_d$ grows proportionally to $\delta$, both in the case of non-linear softening and hardening, including when $z_{rms}$ fully saturates (Fig. 3). This proportionality is expected from elasticity theory with a third order geometrical non-linearity[50] and we experimentally show here that it still holds when symmetry breaking and higher-order non-linearities come into play (Supplementary Note 7).

## Discussion

The large values of $\varepsilon_d \gg \varepsilon_d^h$ reported in Figs. 2–4 cannot be understood as a simple geometrical effect arising from the time-averaged harmonic oscillations of mode profile that remains smooth over the whole drum area. Instead, the enhancement of $\varepsilon_d$ could arise from so-called localisation of harmonics, a phenomenon recently observed in larger and thicker (~500 μm wide, ~500 nm thick) SiN membranes[51] showing RMS displacement saturation similar to Fig. 3a. As the resonator enters the saturation regime, non-linearities (either intrinsic[44], geometrical[50,52] or electrostatically-induced[5,7,53]) may lead to internal energy transfer towards harmonics of the driven mode (Supplementary Fig. 17) and, crucially, to the emergence of ring-shaped patterns over length scales significantly smaller than the size of the membrane[51]. The large displacement gradients associated with these profiles thus enhance $\varepsilon_d$ (Supplementary Note 7). The mode profiles get increasingly complex as the driving force increases, explaining the rise of $\varepsilon_d$ even when $z_{rms}$ reaches a saturation plateau. Considering our study, with $\varepsilon_d \sim 40 \varepsilon_d^h$, we may roughly estimate that large mode profile gradients develop on a scale of $a/\sqrt{40} \approx 500$ nm that is smaller than our spatial resolution (see "Methods" section). Finally, the fact that $\Delta\Gamma_{G,2D}$ (Fig. 2d and Supplementary Fig. 16) is smaller than the associated $\Delta\omega_{G,2D}$ (Figs. 2c and 3a) suggests that the oscillations of $\varepsilon_d(t)$ are rectified

under strong non-linear driving, an effect that further increases the discrepancy between the time-averaged $\varepsilon_d$ we measure and $\varepsilon_d^h$.

Combining multi-mode opto-mechanical tomography and hyperspectral Raman mapping on larger graphene drums (effectively leading to a higher spatial resolution) would allow us to test whether localisation of harmonics occurs in graphene and to possibly correlate this phenomenon to the dynamically-induced strain field. More generally, unravelling the origin of the anomalously large $\varepsilon_d$ may require microscopic models that may go beyond elasticity theory[54] and explicitly take into account the ultimate thinness and atomic structure of graphene[55,56].

Concluding, we have unveiled efficient coupling between intrinsic microscopic degrees of freedom (here optical phonons) and macroscopic non-linear mechanical vibrations in monolayer graphene resonators. Room temperature resonant mechanical vibrations with $\approx 10$ nm RMS amplitude induce unexpectedly large time-averaged tensile strains up to $\approx 4 \times 10^{-4}$. Realistic improvements of our setup, including phase-resolved Raman measurements[35,36] could permit to probe dynamical strain in finer detail, including in the linear regime, where the effective coupling strength[28] could be extracted. For this purpose, larger resonant displacements may be achieved at cryogenic temperatures. In addition, graphene drums, as a prototypical non-linear mechanical systems, can be engineered to favour mode coupling and frequency mixing, which in return can be readout through distinct modifications of their spatially-resolved Raman scattering response.

Our approach can be directly applied to a variety of 2D materials and related van der Waals heterostructures. In few-layer systems, rigid layer shear and breathing Raman-active modes[29,33] could be used as invaluable probes of in-plane and out-of-plane dynamical strain, respectively. Strain-mediated coupling could also be employed to manipulate the rich excitonic manifolds in transition metal dichalcogenides[57], as well as the single photon emitters they can host[58,59]. More broadly, light absorption and emission could be controlled electro-mechanically in nano-resonators made from custom-designed van der Waals hetero-structures[60]. Going one step further, with the emergence of 2D materials featuring robust magnetic order and topological phases[61], that can be probed using optical spectroscopy, we foresee new possibilities to explore and harness phase transitions using nanomechanical resonators based on 2D materials[62,63].

## Methods

**Device fabrication**. Monolayer graphene flakes were deposited onto pre-patterned 285 nm-SiO$_2$/Si substrates, using a thermally assisted mechanical exfoliation scheme as in ref. [64]. The pattern is created by optical lithography followed by reactive ion etching and consists of hole arrays (5 and 6 μm in diameter and 250 ± 5 nm in depth) connected by ~1 μm-wide venting channels. Ti(3 nm)/Au(47 nm) contacts are evaporated using a transmission electron microscopy grid as a shadow mask[34] to avoid any contamination with resists and solvents. Our dry transfer method minimises rippling and crumpling effects[65], resulting in graphene drums with intrinsic mechanical properties (see ref. [31] and Supplementary Note 5 for details). We could routinely obtain pristine monolayer graphene resonators with quality factors in excess of 1500 at room temperature in high vacuum.

**Optomechanical measurements**. Electrically connected graphene drums are mounted into a vacuum chamber ($5 \times 10^{-5}$ mbar). The drums are capacitively driven using the Si wafer as a backgate and a time-dependent gate bias $V_g(t) = V_{dc} + V_{ac} \cos \Omega t$ is applied as indicated in the main text. The applied force is given by $\epsilon_0 \pi a^2 \frac{V_g^2(t)}{2d^2(\xi)}$, where $a$ is the drum radius, $\epsilon_0$ the vacuum dielectric constant, $d(\xi) = (d_{vac} - \xi) + d_{SiO_2}/\epsilon_{SiO_2}$ the effective distance between graphene and the Si substrate, with $\xi$ the static displacement, $d_{vac}$ the graphene-SiO$_2$ distance in the absence of any gate bias, $d_{SiO_2}$ the thickness of the residual SiO$_2$ layer. This force contains a static component proportional to $V_{dc}^2$, which sets the value of $\xi$ and a harmonic driving force proportional to $V_{dc} V_{ac} \cos(\Omega t)$. Note that since $V_{ac} \ll |V_{dc}|$, we can safely neglect the force $\propto V_{ac}^2(1 + \cos(2 \Omega t))$ throughout our analysis.

A 632.8 nm HeNe continuous wave laser with a power of ~0.5 mW is focused onto a ~1.2 μm-diameter spot and is used both for optomechanical and Raman measurements. Unless otherwise stated, (e.g., insets in Figs. 1b and 4), measurements are performed at the centre of the drum. The beam reflected from the Si/SiO$_2$/vacuum/graphene layered system is detected using an avalanche photodiode. In the driven regime, the mechanical amplitude at $\Omega/2\pi$ is readout using a lock-in amplifier. Mechanical mode mapping is implemented using a piezo scanner and a phase-locked loop. For amplitude calibration, the thermal noise spectrum is derived from the noise power spectral density of the laser beam reflected by the sample, recorded using a spectrum analyser. Importantly, displacement calibration is performed assuming that the effective mass of our circular drums is $\tilde{m} = 0.27 \, m_0$ (ref. [46]), with $m_0$ the pristine mass of the graphene drum. As discussed in details in Supplementary Note 5, this assumption is validated by two other displacement calibration methods performed on a same drum. These calibrations are completely independent of $\tilde{m}$. We therefore conclude that to experimental accuracy, our graphene drums are pristine and do not show measurable fingerprints of contamination by molecular adsorbates[66], as expected for a resist-free fabrication process.

**Micro-Raman spectroscopy**. The Raman scattered light is filtered using a combination of a dichroic mirror and a notch filter. Raman spectra are recorded using a 500 mm monochromator equipped with 300 and 900 grooves/mm gratings, coupled to a cooled CCD array. In addition to electrostatically-induced strain, electrostatically-induced doping might in principle alter the Raman features of suspended graphene[34]. Pristine suspended graphene, as used here, is well-known to have minimal unintentional doping ($\lesssim 10^{11}$ cm$^{-2}$) and charge inhomogeneity[66,67]. Considering our experimental geometry, we estimate a gate-induced doping level near $3 \times 10^{11}$ cm$^{-2}$ at the largest $|V_{dc}| = 10$ V applied here. Such doping levels are too small to induce any sizeable shift of the G-mode and 2D-mode features[34,40,66]. In the dynamical regime, the RMS modulation of the doping level induced by the application of $V_{ac}$ is typically two orders of magnitude smaller than the static doping level and can safely be neglected. Similarly, the reduction of the gate capacitance induced by equilibrium position upshifts discussed in Figs. 2e and 3a–iv does not induce measurable fingerprints of reduced doping on graphene.

Let us note that since the lifetime of optical phonons in graphene (~1ps)[68] is more than three orders of magnitude shorter than the mechanical oscillation period, Raman scattering processes provide an instantaneous measurement of $\varepsilon_d$. However, since our Raman measurements are performed under continuous wave laser illumination, we are dealing with time-averaged dynamical shifts and broadenings of the G-mode and 2D-mode features. Raman G-mode and 2D-mode spectra are fit using one Lorentzian and two modified Lorentzian functions, as in refs. [31,67], respectively (Supplementary Note 1). As indicated in the main text, Grüneisen parameters of $\gamma_G = 1.8$ and $\gamma_{2D} = 2.4$ are used to estimate $\varepsilon_s$ and $\varepsilon_d$. These values have been measured in circular suspended graphene blisters under biaxial strain[31]. Considering a number of similar studies[31,32,34,49,69], we conservatively estimate that the values of $\varepsilon_s$ and $\varepsilon_d$ are determined with a systematic error lower than 20%. Such systematic errors have no impact whatsoever on our demonstration of dynamically-enhanced strain. Finally, the Raman frequencies and the associated $\varepsilon_s$ and $\varepsilon_d$ are determined with fitting uncertainties represented by the errorbars in the figures.

## Data availability

The datasets generated during and/or analysed during this study are available from the corresponding authors (X.Z. and S.B.) on reasonable request.

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

## Acknowledgements

We thank T. Chen, A. Gloppe and G. Weick for fruitful discussions. We thank the StNano clean room staff (R. Bernard and S. Siegwald), M. Romeo, F. Chevrier, A. Boulard

and the IPCMS workshop for technical support. This work has benefitted from support provided by the University of Strasbourg Institute for Advanced Study (USIAS) for a Fellowship, within the French national programme "Investment for the future" (IdEx-Unistra). We acknowledge financial support from the Agence Nationale de Recherche (ANR) under grants H2DH ANR-15-CE24-0016, 2D-POEM ANR-18-ERC1-0009, as well as the Labex NIE project ANR-11-LABX-0058-NIE.

## Author contributions

The project was originally proposed by S.B and P.V (GOLEM project, supported by USIAS). K.M. and D.M. built the experimental setup, with help from X.Z. X.Z. fabricated the samples, with help from H.M., D.M. and K.M. X.Z. carried out measurements, with help from K.M. and L.C. X.Z. and S.B. analysed the data with input from K.M., L.C. and P.V. X.Z. and S.B. wrote the manuscript with input from L.C. and P.V. S.B. supervised the project.

## Competing interests

The authors declare no competing interests.
