## [Peer Review File · Nature Communications]

Reviewer #1 (Remarks to the Author):

In Zhang et al., the authors used their previously established micro-Raman scattering spectroscopy technology [reported in 2D Mater. 4, 014004 (2016), as Ref. [41]] to study time-averaged Raman spectra in graphene-based nanomechanical resonator under strong drive in the nonlinear regime. By measuring the shift, broadening, and intensity of G-mode and 2D-mode features, they extracted strain induced by electrostatic gating and strong microwave driving (they refer as 'dynamically-induced strain'). And they mapped dynamically-induced strain's dependency on driving frequency and position.

I find the novelty of the manuscript needs to be clearly stated and clarified in more detail, over their previous work (Ref. [41]), since the experimental and analyzing methods are quite similar.

Meanwhile, the authors attribute the observed unexpected large strain originated from dynamical enhancement, in which sense, the doping effect is excluded. They estimate the Dirac point of the device to be around 0.75 V from the 'U-shape' frequency-tuning spectrum, while the symmetric gate dependence is actually due to electrostatic force induced by gate voltage rather than transport properties. Also, the static deflection and vibration amplitude may lead to the change of gate capacitance thus affecting the doping level as well. The authors need to carefully check the contributions from doping effect.

In this sense, the manuscript does not meet the criterion of Nature Communications and I cannot recommend its publication.

Reviewer #2 (Remarks to the Author):

The manuscript reports nanomechanical studies of monolayer graphene drums. The drums are made by suspending graphene over holes made in SiO₂ on Si wafers and are capacitively actuated. The mechanical motion is detected using optical interferometric technique. This aspect is well established in the literature.

The main part of the work is the use of Raman spectroscopy to track the strain in the drums. The authors report simultaneous Raman measurement while the drum is oscillating. By tracking the Raman modes (G and 2D) the strain is "extracted" by measuring shifts in Raman modes. The "calibration" of strain and Raman mode shift is done by static deflection using the DC gate voltage via modelling of deflection and static Raman shift. The idea of using Raman modes to probe strain in nanomechanical devices is well established even before the arrival of 2D materials.

The novel result of the paper is the large $\sim 4 \times 10^{-4}$ "dynamically enhanced" strain. This strain is induced is due to the anharmonic nature of the potential of the resonator which results in a new equilibrium position – leading to strain. It should be noted that anharmonic nature of the potential is due to the electromechanical potential (with contributions of the nonlinear electrostatic potential) and not just purely mechanical/elastic contribution.

While the measurements are interesting and the data is comprehensive, the disconnect as the authors describe "These anomalously large strains exceed the time-averaged values predicted for harmonic vibrations with the same root mean square (RMS) amplitude by more than one order of magnitude" does not make sense. It is plausible but not shown numerically by authors – in fact the authors do not provide a clear idea as to why this should happen?

At this point the manuscript does not address all aspects of the system. If the authors address the following questions convincingly then I will be happy to recommend publication of the manuscript in Nature comm. I want to give the authors an opportunity to respond to the questions.

1. If the actuation scheme was thermomechanical (using another laser) would they see the same levels of strain generation?
2. In Fig.2(c) inset, the slope doesn't seem to be 2.2 but a bit less than that.
3. How sensitive is the shift in a Raman peak on the deviation of drive signal frequency from the backbone curve?
4. In Fig.3(a), is there any simple way to explain the fact that tensile strain is increasing but the r.m.s. amplitude of vibration is saturated before the jump?
5. Is there any physical way to understand why ϵ_d is proportional to Δ ?
6. Is the frequency shift to strain calibration model dependent?
7. In how many devices do they see this ? Please present data from all devices in SI.
8. The explanation for this observation should be substantial.

Reviewer #3 (Remarks to the Author):

Review of: Dynamically-enhanced strain in atomically thin resonators

The authors describe an interesting experimental study of the strain in a resonating graphene membrane. For this purpose they employ for the first time a simultaneous measurement of the Raman phonons and the mechanical motion of the membrane. Interestingly they find evidence that the strain in the graphene increases at higher driving forces.

The experimental results are new and relevant and certainly worth publication, however I have some difficulties with the presented analysis of the data that needs to be addressed before publication, since the conclusions might be inaccurate. In particular the conclusion: "These

anomalously large strains exceed the time-averaged values predicted for harmonic vibrations with the same root mean square (RMS) amplitude by more than one order of magnitude” needs stronger backup.

I would like the authors to address these comments:

1. When a membrane resonates around a flat equilibrium position $z_0=0$, I agree with the authors that the time-averaged strain increases with driving amplitude as indicated by equation (S5), because both for positive and negative deflections the strain increases.

However, when the membrane resonates around a bent driving amplitude, with z_0 not equal to zero (and $z_0 > \sqrt{2} z_{rms}$), $z(t)=z_0+\sqrt{2}z_{rms}\cos(\omega t)$ and it is not so obvious why the average strain in the membrane increases. The membrane will move around a certain equilibrium point in Fig. S6 and its strain will increase half of the time (w.r.t. the average strain), but be lower than the average strain the other half of the time.

It is not clear to me, why under these conditions, the dynamic motion will significantly increase average strain. I would only expect a broadening of the Raman peak without a significant shift in the average value. This broadening is also observed and might contain the most important information on the strain fluctuations, but is not analysed in as much detail.

2. Equation (S5) seems only valid for the flat membrane configuration. For a membrane that is bent by electrostatic force induced by V_{dc} , the effect of z_{rms} on strain is roughly indicated by the slope of the curve in Fig. S6, so the induced strain will not only depend on z_{rms} but also on the value of V_{dc} (and z_0).

Like in the previous comment, the broadening of the peak is expected to depend on the product of the slope of the curve in Fig. S6 and z_{rms} .

The use of equation (S5) to estimate the expected strain increase seems not correct and can explain why the authors come to conclusion that “These anomalously large strains exceed the time-averaged values predicted for harmonic vibrations with the same root mean square (RMS) amplitude by more than one order of magnitude”, that might be erroneous.

3. It has been reported that the strain measured by Raman is different from the strain measured from the displacement of the graphene membrane due to hidden area effects in this work:

<https://journals.aps.org/prl/abstract/10.1103/PhysRevLett.118.266101>

Since the authors also compare these two methods to analyse strain of graphene, it would be of interest to the reader if the authors can comment whether they see a similar discrepancy between strain estimated from membrane deflection and strain estimated from Raman.

4. The three comments above are the key points. Some more minor comments:

a. There have been discrepancies in the definition in of the 2D Young’s modulus of graphene. From 3D mechanics it is found that: the ration of the biaxial tension and biaxial strain is given by: $T_{biaxial}/\epsilon_{biaxial} = E_{3D}t/(1-\nu)$. Where E_{3D} is the 3D Young’s modulus, t the membrane thickness and ν the Poisson ratio of an isotropic material. At the start of page 4 of the manuscript it is given that $T_0/\epsilon_{s0} = E_{1LG}/(1-\nu^2)$. That E_{1LG} doesn’t seem to agree with the common definition $E_{2D} = E_{3D}t$, can the authors explain why?

b. The authors explain that since $V_{dc}^2 \gg V_{ac}^2$, and because symmetry breaking effects are small, changes in the equilibrium position and nonlinearity at higher V_{ac} are small. However, nonlinear electrostatic force effects can also arise because the condition $z_{rms} \ll d$ is violated ($F_{el} = A \epsilon_0 V^2/[2(d-z)^2]$). Since gap $d=250$ nm and initial deflection due to the DC voltage is 60 nm, and z_{rms} is of the order of 10 nm this inequality might not hold in. The authors are encouraged to comment if these electrostatic nonlinearities are relevant.

Reply to Reviewer #1

Original report by Reviewer #1:

In Zhang et al., the authors used their previously established micro-Raman scattering spectroscopy technology [reported in 2D Mater. 4, 014004 (2016), as Ref. [41]] to study time-averaged Raman spectra in graphene-based nanomechanical resonator under strong drive in the nonlinear regime. By measuring the shift, broadening, and intensity of G-mode and 2D-mode features, they extracted strain induced by electrostatic gating and strong microwave driving (they refer as ‘dynamically-induced strain’). And they mapped dynamically-induced strain’s dependency on driving frequency and position. I find the novelty of the manuscript needs to be clearly stated and clarified in more detail, over their previous work (Ref. [41]), since the experimental and analyzing methods are quite similar. Meanwhile, the authors attribute the observed unexpected large strain originated from dynamical enhancement, in which sense, the doping effect is excluded. They estimate the Dirac point of the device to be around 0.75 V from the ‘U-shape’ frequency-tuning spectrum, while the symmetric gate dependence is actually due to electrostatic force induced by gate voltage rather than transport properties. Also, the static deflection and vibration amplitude may lead to the change of gate capacitance thus affecting the doping level as well. The authors need to carefully check the contributions from doping effect. In this sense, the manuscript does not meet the criterion of Nature Communications and I cannot recommend its publication.

Our reply - We thank the reviewer for considering our manuscript and are sorry that he/she “cannot recommend its publication” primarily because in her/his opinion i) our work bears too much similarity with our previous study (by Metten *et al.*, listed as Ref 41 in the submitted manuscript, now listed as Ref. 34) and ii) we do not sufficiently consider doping effects in our analysis. Let us mention directly that we beg to differ with the Reviewer’s statements, which we are afraid arise from overlooking the conclusions of our previous works and of our current submission. There is a major improvement between our 2D Materials paper (i.e., Ref 34) that focuses on static effects and this more advanced work on dynamically-induced effects. We have strived to make this point as clear as possible in our revised version. In the following, we provide replies to the Reviewer’s comments that we have extracted from her/his report.

RC1.1 - *I find the novelty of the manuscript needs to be clearly stated and clarified in more detail, over their previous work (Ref. [41]), since the experimental and analyzing methods are quite similar.*

AR1.1 - we would kindly like to mention that this comment by the reviewer is, in opinion, a bit of a judgement call. We hope that our detailed reply will convince her/him to reconsider his recommendation on our study. The reviewer is right that Ref 41 combines electrostatic gating and Raman spectroscopy. However, this is to investigate suspended graphene samples only in the static regime. The same techniques are used in our manuscript **in the dynamical regime**. We have done a major experimental effort to combine, for the first time, state of the art opto-electro-mechanical measurements on pristine graphene resonators with state of the art Raman spectroscopy. This new approach provides entirely new insights into dynamically-induced (and in fact “enhanced”) strain. Our approach is original and timely; it is both fundamentally and technologically relevant for it address systems with unique opto-electro-mechanical properties that hold great promise for hybrid opto-electro-mechanics and nanoscale sensing. Hence, our work is undeniably a leap forward compared to Ref. 41 and also compared to previous opto-mechanical studies in resonators made from graphene and related systems.

CM1.1 - The sentence “Recently, the interplay between electrostatically-induced strain and doping has been probed in the static regime in suspended graphene monolayers~\cite{Metten2016}.” has been added in the introduction. This sentence is followed by “Dynamically-induced strain has been investigated using Raman spectroscopy in bulkier micro electro-mechanical systems, including mesoscopic graphite cantilevers but remains unexplored in resonators made from 2D materials.”

RC1.2 - They estimate the Dirac point of the device to be around 0.75 V from the ‘U-shape’ frequency-tuning spectrum, while the symmetric gate dependence is actually due to electrostatic force induced by gate voltage rather than transport properties.

AR1.2 - The reviewer is right that the “U-shaped” curve is due to the electrostatic force that increases the tension in our membrane and hence its mechanical resonance frequency, as extensively established in the literature (see Refs 4 to 8). The electrostatic force vanishes when the sample is neutral such that we expect a symmetric behavior with respect to the charge neutrality point, as we observe in experiments. The charge neutrality point is, in our work at 0.75 V, which translates into a very weak native doping level of less than $2 \cdot 10^{10} \text{ cm}^{-2}$. Our point here is also that suspended graphene samples are quasi neutral and immune to unintentional doping, as we have intensively documented in our previous papers.

CM1.2 - We have added the following sentence on page 3: “From the value of V_{dc}^0 , we estimate a minute unintentional hole doping below $2 \times 10^{10} \text{ cm}^{-2}$ that is consistent with the pristine character of suspended graphene~\cite{Berciaud2009,Berciaud2013}.”

RC1.3 - Also, the static deflection and vibration amplitude may lead to the change of gate capacitance thus affecting the doping level as well. The authors need to carefully check the contributions from doping effect.

AR1.3 - Our work by Metten *et al.* (now Ref. 34 in our revised manuscript) studies quantitatively how electrostatic gating can lead to strain and (as the reviewer suggests) doping in suspended graphene samples. In this reference, we have investigated these effects in the static regime, where large enough levels of static strain up to $\sim 10^{-3}$ and measurable doping (at most $6 \times 10^{11} \text{ cm}^{-2}$ near sample collapse) can be attained. Strain and doping are quantitatively assessed through Raman scattering measurements (as in our submission) and compared to a detailed electromechanical model where the displacement-dependent gate capacitance is considered. **We have come to the unambiguous conclusion that the effects of doping on the Raman fingerprints are negligible** and at most secondary near device collapse. Indeed, it is well established that the Raman features of graphene are only sensitive to doping levels above $\approx 5 \times 10^{11} \text{ cm}^{-2}$ (see for example Fig. 4 in Ref. 34). These doping levels are just attained (again, near device collapse) in Ref. 34 and are not explored in the present study, where we can estimate a doping level slightly below $3 \times 10^{11} \text{ cm}^{-2}$ for the largest gate bias $V_{\text{dc}} = 10 \text{ V}$ applied in our submitted manuscript. Importantly, the doping level will only be marginally affected with the resonant motion since $V_{\text{ac}} \ll V_{\text{dc}}$ and the main consequence of doping in our measurements is the electrostatic damping that we mention in the main text, an effect that is well-known to the nanomechanics community.

Fig. 4d of Ref. 34 shows that by performing a careful analysis of the Raman G-mode linewidth, we may indeed sense the effect of doping, which induces a reduction of the G-mode linewidth, well studied in the literature (see refs 40), including by us (Ref. 42). Conversely, electrostatically induced static strain can approach 10^{-3} , an amount of strain which leads to easily measurable Raman shifts of the G and 2D modes that largely dominate all possible Raman shifts and linewidth changes due to the slight doping induced by the gate bias. This is clearly evidenced by the large $\frac{\partial \omega_{2D}}{\partial \omega_G} \approx 2.2$ slope which exceeds the well-known values

expected under electron or hole doping that are well below 1 (see reference 42 for details). This point is already mentioned in our submitted manuscript. For the sake of completeness, we have strengthened this claim and referred more extensively to our Refs. 34, 42 and 45 (see changes made).

CM1.3 -

Main text, Page 3, top of 1st column - We have added the sentence *“This difference allows a clear disambiguation between strain and doping (see Methods for details).”*

Methods - We have added the following paragraph we refer to it in the main text. *“In addition to electrostatically-induced strain, electrostatically-induced doping might in principle induce changes in the Raman features of suspended graphene~\cite{Metten2016}. Considering our experimental geometry, we estimate a gate-induced induced doping level near $3 \times 10^{11} \text{ cm}^{-2}$ at the largest $V_{dc} = 10 \text{ V}$ applied here. Such doping levels are too small to induce any sizeable shift of the G- and 2D-mode features~\cite{Pisana2007,Berciaud2009,Metten2016}. In the dynamical regime, the RMS modulation of the doping levels induced by the application of V_{dc} is typically two orders of magnitude smaller than the static doping level and can safely be neglected.”*

References 41, 40 and 61 in the original manuscript now appear as Refs 34 42 and 45 in our revised manuscript, respectively.

Reply to Reviewer #2

Original Report by Reviewer #2:

The manuscript reports nanomechanical studies of monolayer graphene drums. The drums are made by suspending graphene over holes made in SiO₂ on Si wafers and are capacitively actuated. The mechanical motion is detected using optical interferometric technique. This aspect is well established in the literature.

The main part of the work is the use of Raman spectroscopy to track the strain in the drums. The authors report simultaneous Raman measurement while the drum is oscillating. By tracking the Raman modes (G and 2D) the strain is “extracted” by measuring shifts in Raman modes. The “calibration” of strain and Raman mode shift is done by static deflection using the DC gate voltage via modelling of deflection and static Raman shift. The idea of using Raman modes to probe strain in nanomechanical devices is well established even before the arrival of 2D materials.

The novel result of the paper is the large $\sim 4 \times 10^{-4}$ “dynamically enhanced” strain. This strain is induced is due to the anharmonic nature of the potential of the resonator which results in a new equilibrium position – leading to strain. It should be noted that anharmonic nature of the potential is due to the electromechanical potential (with contributions of the nonlinear electrostatic potential) and not just purely mechanical/elastic contribution.

While the measurements are interesting and the data is comprehensive, the disconnect as the authors describe “These anomalously large strains exceed the time-averaged values predicted for harmonic vibrations with the same root mean square (RMS) amplitude by more than one order of magnitude” does not make sense. It is plausible but not shown numerically by authors – in fact the authors do not provide a clear idea as to why this should happen?

At this point the manuscript does not address all aspects of the system. If the authors address the following questions convincingly then I will be happy to recommend publication of the manuscript in Nature comm. I want to give the authors an opportunity to respond to the questions.

Our general reply - We would like to thank the reviewer for her/his careful reading of our manuscript and her/his encouragements to address her/his comments in a revised version before publication in nature communications. The reviewer acknowledges the novelty of our work and urges us to provide more microscopic insights to rationalize the dynamically-enhanced strain we have unveiled for the first time. In the following we first bring some elements of clarification following parts of her/his reports. We then proceed and address the reviewer’s comments in full, on a point by point basis.

First of all, we would kindly like to comment on two statements in the Reviewer’s report. The Reviewer wrote:

Reviewer Statement #S1 - *The “calibration” of strain and Raman mode shift is done by static deflection using the DC gate voltage via modelling of deflection and static Raman shift.”*

AR#S1 - One should distinguish between static strain and static deflection. The Raman frequency of the G and 2D modes provide a direct determination of biaxial static strain. Indeed, the determination of strain is straightforwardly related to the well-established Grüneisen parameters in graphene.

In contrast, the graphene deflection has to be calibrated. In the static regime, the deflection is determined from the levels of static biaxial strains inferred by Raman measurements, assuming a parabolic deflection

profile in the static regime. The deflection could also be obtained interferometrically, based on the Raman scattering intensity, as in Ref 34 or Ref. 31 (Metten *et al.*, Phys. Rev Applied, 2014), or in a more straightforward manner using the Gate-dependent reflectivity.

In conclusion, there is no ambiguity regarding the calibration of strain since, again, Raman scattering is a “calibration free” method. There may however be difficulties in calibrating the mechanical RMS displacement. Aware of these difficulties, we have, in our comprehensive study, provided extensive details on our displacement calibration methods (Section S5) and established displacement calibration based on three complementary and consistent methods. Such a calibration is essential to support our claim of dynamically-enhanced strain.

Reviewer Statement 2 - *“This strain is induced is due to the anharmonic nature of the potential of the resonator which results in a new equilibrium position – leading to strain.”*

AR#S2 - We could like to clarify this statement. In fact, we demonstrate in our work that Raman scattering measurements make it possible to determine dynamical equilibrium position shifts, which is an important scientific result in itself shifts (see paragraph entitled: Equilibrium position shift in page 4 and Fig. 2e and Fig. 3a-iv). We do also stress that dynamical equilibrium shifts alone **cannot account for the tensile dynamical strain we observe** in our manuscript (see also our new supplementary section S7).

CM#S2 - The reviewer comment urges us to clarify the disconnection between equilibrium position shift and dynamically enhanced strain. In our revised manuscript, we have added the sentence *“From these measurements, we conclude that the dynamical softening of ω_G and ω_{2D} is not due to an equilibrium position shift.”* to close the section entitled *“Equilibrium position shift”*.

RC2.1 - *If the actuation scheme was thermomechanical (using another laser) would they see the same levels of strain generation?*

AR2.1 - We thank the reviewer for this very relevant question. We are aware that driving the system optically may allow us to pinpoint the share of electrostatically-induced non-linear effects in our study and in particular demonstrate if dynamically-enhanced strain also occurs in the absence of electrostatically-induced mechanical non-linearities.

It is generally quite challenging to achieve well-controlled and large RMS displacements using thermomechanical (or photothermal) actuation. Under strong optical driving, photothermal backaction may lead to sample-specific non-linear regimes of parametric amplification and self-oscillation (See Barton et al., cited as Ref. 10). Using excessive laser powers would also lead to sample damage and mechanical instabilities associated with relatively slow heat transport (on time scales possibly longer than the period of the mechanical oscillations) without the possibility to monitor the Raman response reliably.

For these reasons we have privileged electrostatic actuation in our study. As indicated in the rest of this rebuttal, we claim that the large levels of strain attained here stem from geometrical non-linearities in the sense that geometrical non-linearities are directly linked to the level of strain in the drum (see our new Supplementary section S7). Whether and how such non-linearities can be generated through photothermal actuation will be the focus of future research. We would like to stress that the focus in this work is to demonstrate that Raman scattering spectroscopy can act as a very sensitive probe of complex

strain fields (observed here in the non-linear regime) and that, conversely, electrostatically-induced strain can be used to modulate intrinsic quantum degrees of freedom (here phonons) in graphene.

For the sake of completeness let us mention photothermal actuation is an interesting approach to explore intriguing backaction effects and also tune mechanical non-linearities. These effects will certainly be the focus of further studies in the coming years. Nevertheless, the shortcomings associated with photothermal actuation (heating, self-oscillations, mechanical instabilities) would make it very challenging to perform the comprehensive study reported in our paper.

CM2.1 - On page 2 of our revised manuscript, we have added the following sentence to justify our choice of actuation scheme. *“We have chosen electrostatic rather than photothermal actuation\cite{Sampathkumar2006} to attain large RMS amplitudes while at the same time avoiding heating and photothermal backaction effects~\cite{Barton2012,Morell2019}, possibly leading to self-oscillations, mechanical instabilities and sample damage.”.*

As Ref 39, We now cite A. Sampathkumar *et al.*, APL **88**, 223104 (2006) as an early work that employed photothermal actuation in nanomechanical resonators.

RC2.2 - In Fig.2(c) inset, the slope doesn't seem to be 2.2 but a bit less than that.

AR2.2 - This point is related to comment RC6 and also partially addressed in our reply AR6. The reviewer is right that we indeed see a deviation from a 2.2 slope at the largest driving forces employed in this study. This effect may stem from the slight broadening of the 2D mode feature observed under strong driving (Fig. 2e) and was not systematically observed in our experiments (see for instance Fig. 3c).

In our study, we have considered Grüneisen parameters that were measured on similar systems, namely pressurized graphene blisters (Ref. 31) and the resulting slope of 2.2 is shown as a guide to the eye and not as a fit, which indeed would lead to a slightly smaller slope.

There is a scatter among the reported Grüneisen parameters in the literature (see comment RC6 by the reviewer). Still, the key point is that the $\frac{\partial\omega_{2D}}{\partial\omega_G}$ slope observed here for the first time in a driven graphene drum is very similar to that obtained in the static regime (see Fig. 1d and also reference 34 on other samples) and close to the 2.2 value that indicates biaxial strain. As indicated in our manuscript and also in our reply to Reviewer 1, this slope much larger than the $\frac{\partial\omega_{2D}}{\partial\omega_G}$ slope observed in the case of electron- or hole-doped graphene (see our Ref. 42 for details).

For the sake of completeness, let us stress that, as stated in our manuscript, the 2D mode frequency plotted in figure 2 is the frequency of the low energy 2D mode subfeature (ω_{2D-} see Sec. S1.b in the SI file for details). The frequency ω_{2D+} of the higher energy subfeature is shown (with an offset, for clarity) below along with ω_{2D-} as a function of ω_G . The 2D⁺ feature follows a slope closer to 2.2.

Figure 1- Frequencies of the $2D^{\pm}$ subfeatures (see Sec S1b for details) as a function of ω_G for the data in figure 2. The $2D^+$ feature is rigidly offset for clarity. The solid line is a guide to the eye with a slope of 2.2.

CM2.2 - We are now showing the above figure in our SI file (listed as Fig. S4) accompanied with a brief discussion. We refer to this figure in our manuscript (bottom on page 3, column 2).

RC2.3 - How sensitive is the shift in a Raman peak on the deviation of drive signal frequency from the backbone curve?

AR2.3 - Along the backbone curve, the mechanical resonance frequency shifts non-linearly with the RMS amplitude (as indicated in section S6b, Eq. (S17)). The reviewer rightly wonders how sensitive the dynamically-induced Raman shift (or equivalently dynamically-induced strain ϵ_d) is relative to the RMS amplitude. Or, in other words, how accurately Raman scattering may allow us to monitor the backbone curve? As shown in Fig. 2f and 3d and further detailed in our revised manuscript, SI file (Sec. S7) and reply to comment RC2.5 by the Reviewer, we establish experimentally and theoretically the proportionality between non-linear frequency shift and ϵ_d .

CM2.3 - We have added a new section S7 in the SI file entitled (Dynamical strain and non-linearities). Please also see changes made related to comment RC5.

RC2.4 - In Fig.3(a), is there any simple way to explain the fact that tensile strain is increasing but the r.m.s. amplitude of vibration is saturated before the jump?

AR2.4 - In our manuscript we stress that we measure the Fourier component of the RMS amplitude at the drive frequency, which may fully saturate in the non-linear regime, as in Fig. 3. If one considers a Duffing-like third order non-linearity (should it have an intrinsic, electrostatic or geometrical origin), no amplitude saturation should be expected as the driving amplitude is linearly increasing (see Ref. 47 in our revised manuscript). As the resonator is driven further into the nonlinear regime, higher-order nonlinearities appear leading to internal energy transfer between eigenmodes of the resonator or between harmonics of the fundamental oscillation. Such coupling leads to non-trivial mode profiles, as shown for instance in Ref. 52 by F. Yang *et al.* in the case of bulkier SiN membranes. These mode profiles get increasingly complex as the driving force increases leading to larger dynamical strain (see our new Section S7 in the

revised SI file). At the same time, the total RMS amplitude (including all Fourier components) saturates but the non-linear mechanical frequency shift δ may increase further.

We would like to stress that only very few reports on the strongly-nonlinear mechanical response of NEMS and MEMS were reported in the literature, due to the difficulty in achieving large displacements in resonators that are bulkier than our atomically thin systems (see for instance <https://iopscience.iop.org/article/10.1088/0960-1317/20/10/105012>). Non-linear oscillations in clamped membranes have recently attracted more attention and this topic is now timely. We do cite ref. Ref. 52 and another theoretical preprint by the group of E. Collin in Grenoble (Ref 54).

In our work, atomically thin resonators are put forward as strongly non-linear mechanical systems, allowing the observation of large non-linear frequency shifts (denoted δ in our manuscript) and dynamically-enhanced strain. This point is further discussed in our reply to comments RC5 and RC8 by the reviewer.

CM2.4 - We now cite Ref. 54 mentioned above and have added the following text in the discussion section (see revised MS for associated references).

“As the resonator enters the saturation regime, non-linearities (either intrinsic, geometrical or electrostatically-induced) may lead to internal energy transfer towards harmonics of the driven mode (Fig. S17) and, crucially, to the emergence of ring-shaped patterns over length scales significantly smaller than the size of the membrane. The large displacement gradients associated with these profiles thus enhance ϵ_d (Supplementary Section S7). The mode profiles get increasingly complex as the driving force increases, explaining the rise of ϵ_d even when z_{rms} reaches a saturation plateau. Considering our study, with $\epsilon_d \sim 40 \epsilon_d^h$, we may roughly estimate that large mode profile gradients develop on a scale of $a\sqrt{40} \approx 500 \text{ nm}$ that is smaller than our spatial resolution (see Methods).”

RC2.5 - Is there any physical way to understand why ϵ_d is proportional to δ ?

AR2.5 - We thank the reviewer for this very relevant question and acknowledge that this essential point deserves further clarification. In our revised manuscript and SI file, we demonstrate that using an elementary mechanical model, the dynamically-induced strain experienced by a vibrating membrane subject to third order geometrical non-linearities is directly proportional to the non-linear frequency shift of the mechanical mode (labelled δ in page 7 of our revised manuscript and also expressed in equation S17 of our SI file). We discuss how this proportionality can still hold when other sources of non-linearity come into play. Please see also our reply to Comment #8 by the reviewer.

CM2.5 -

We have added a new Section (labelled S7, Dynamical strain and non-linearities) in our revised SI file to present this calculation.

We now discuss the proportionality between δ and ϵ_d in the main text. The sentence: *“This result indicates that the enhancement of ϵ_d originates from the strong non-linear driving of the graphene drum and that δ can be regarded as an indirect measurement of ϵ_d .”* has been replaced by: *“This proportionality is expected from elasticity theory with a third order geometrical non-linearity and we experimentally show here that it still holds when symmetry breaking and higher-order non-linearities come into play (see Supplementary Section S7 for details).”* The discussion added as CM4 is also relevant to reply to RC5.

-We are now citing S. Schmid, L. G. Villanueva, and M. L. Roukes, *Fundamentals of nanomechanical resonators* (Springer, 2016) as Ref 53.

RC2.6 - Is the frequency shift to strain calibration model dependent?

AR2.6 - We thank the reviewer for this question. The Raman shift is straightforwardly connected to the biaxial strain via the Grüneisen parameters associated with the G- and 2D- mode features (see the review cited as Ref.29). In the literature, there is a slight scatter in the experimentally determined Grüneisen parameters (see Ref. 32), which mostly arises from the distinct sample geometries used in experiments. We took the slope of $\frac{\partial\omega_{2D}}{\partial\omega_G} = 2.2$ and Grüneisen parameters of 2.4 and 1.8 for the 2D mode and for the G mode, respectively as in Ref. 31 by Metten *et al.* (the corresponding strain-induced shift rates are near $130\text{ cm}^{-1}/\%$ strain and near $60\text{ cm}^{-1}/\%$ strain, for the 2D and G-modes, respectively). These values have been measured in clamped circular graphene drums under biaxial strain, i.e; the same configuration as in our manuscript. As shown in the attached figure below, similar slopes in the range $\frac{\partial\omega_{2D}}{\partial\omega_G} \approx 2.2 - 2.4$ have been consistently reported by others in the same geometry as in our manuscript. Of course, the choice of Grüneisen parameters leads to a possible systematic error in the strain. We estimate that this error is lower than 20%, considering the figure below as well as the results compiled in Ref. 32. Such uncertainties have no impact whatsoever on our demonstration of dynamically-enhanced strain. For the sake of completeness, let us mention that Raman frequencies and associated ϵ_s and ϵ_d are determined with fitting uncertainties represented by the errorbars in the figures of our main manuscript and SI.

Figure 1 - Correlation between the Raman 2D and G mode frequencies in graphene blisters under biaxial tensile strain. “Zabel” refers to Zabel *et al.*, *Nano Letters* **12**, 617 (2012). “Lee” refers to Lee *et al.* *Nano Letters* **12**, 4444, 2012 and “Metten” refers to ref 31, Metten *et al.*, *Phys. Rev. Applied* **2**, 054008 (2014)

CM2.6 - We have added the following text in the Methods section: “These values [of the Grüneisen parameters] have been measured in circular suspended graphene blisters under biaxial strain. Considering a number of similar studies, we conservatively estimate that the values of ϵ_s and ϵ_d are determined with a systematic error lower than 20 %. Such systematic errors have no impact whatsoever on our

demonstration of dynamically-enhanced strain. Finally, Raman frequencies and associated ϵ_s and ϵ_d are determined with fitting uncertainties represented by the errorbars in the figures."

RC2.7 - *In how many devices do they see this? Please present data from all devices in SI.*

AR2.7 - As indicated in the main text we have studied three devices and systematically observed dynamically-enhanced strain in all of them. All the relevant data is already shown in the originally submitted SI file, which contains 20 figures. In particular, our main text and fig S1-S18 focus on devices 1 and 2 and Fig 19-20 report data on device 3. The device name is indicated in each figure of the SI file.

CM2.7 - The sentences "*Complementary and consistent results are reported in Supplementary Section 8 or device 1 and in Supplementary Sections 9 and 10 for other graphene drums with similar designs, denoted devices 2 and 3, respectively. In device 3, we have measured ϵ_d up to $\approx 4 \times 10^{-4}$ for $z_{rms} \approx 14$ nm.*" has been edited into: "*We have consistently observed dynamically-enhanced strain in three graphene drums with similar designs, denoted device 1,2,3. Complementary results are reported in Supplementary Section 9 for device 1 and in Supplementary Sections 10 and 11 for devices 2 and 3, respectively. In device 3, we have measured ϵ_d up to $\approx 4 \times 10^{-4}$ for $z_{rms} \approx 14$ nm.*"

RC2.8 - *The explanation for this observation should be substantial.*

AR2.8 - We address this comment in part in our replies AR2.4 and AR2.5.

We assume that RC2.8 refers to the title of our paper and to our claim of dynamically-enhanced strain. Looking at the derivation of Eq. (S22) in our revised SI file, the strain is defined as an average over the square of the spatial derivative of the mode profile. We can then rationalize the experimentally observed enhancement of ϵ_d as the emergence of an increasingly complex mode profiles in the heavily non-linear regime. Following temporal and spatial averaging, such mode profiles yield a finite net tensile strain that we experimentally measure using Raman spectroscopy as a spectral downshift of the G- and 2D-mode features. The enhanced character of the measured ϵ_d arises from the strong non-linearities (either intrinsic, geometrical or electrostatically-induced) of our resonators and for a given RMS amplitude, ϵ_d largely exceeds the strain induced by temporally-averaged harmonic oscillations (with the same RMS amplitude) of a clamped circular drum with a smooth parabolic profile (see Eq. (S18) in our revised SI file, listed as Eq. (S5) in our initial submission) .

We shall now comment on the absolute value of δ which is as large as 5% (see Fig. 2f) in our pristine graphene drums. Ref 52, PRL **122**, 154301 (2019) by F. Yang *et al.* reports non-linear mechanical response of bulkier systems (namely square SiN drums with > 100 μm in lateral size) that are qualitatively similar to our observations in Fig 2 and 3. This work demonstrates that under strong non-linear driving, the mechanical mode profile of SiN drums develops anomalous shapes with ring patterns that are unambiguously resolved using diffraction-limited optical interferometric readout (see fig. 2a in ref 52). These findings support the idea that geometrical nonlinearities induce complex patterns that enhance dynamical strain, since the strain qualitatively scales $\left(\frac{z_{rms}}{a'}\right)^2$, where a' is the characteristic spatial extension of the mode patterns, with $a' < a$, the typical size of the drum. Noteworthy, these observations are made in a regime where $\delta \lesssim 10^{-3}$, i.e. about 50 times less than in our atomically-thin drums. Along

these lines, we can expect similar spatial patterns in our non-linearly driven graphene drums, likely with a larger number of rings. Considering the small size of our drums (5 or 6 μm in diameter, depending on the device), such patterns will not be appropriately resolved using diffraction limited techniques. Indeed, we measure ϵ_d that about 40 times larger than ϵ_d^h , the value expected under harmonic oscillations. As a rough estimate, this suggests that patterns develop over a length scale smaller than $\frac{a}{\sqrt{40}} \sim 500 \text{ nm}$. This upper bound is thus below our diffraction-limited resolution ($\gtrsim 1 \mu\text{m}$) such that we are presently not able to accurately resolve dynamical strain patterns. As suggested in our original manuscript, future developments will focus on spatially-resolved studies (as in Fig. 4) on larger-diameter graphene drums and hence bridge the gap between the intriguing non-linear geometries in SiN micro-mechanical drums (Ref. 52) and the stronger non-linearities revealed here in the atomically thin limit.

CM2.8 -

Abstract - We have added the following sentence: *“We experimentally demonstrate that the dynamical strain is proportional to the non-linear frequency shift of the mechanical resonance. This key observation is rationalized using a non-linear mechanical model and allows us to attribute the enhancement of dynamical strain to the strong non-linearities of graphene drums.”*

Main text - We have edited and expanded the sections *“Dynamically-enhanced strain”* and *“Discussion”* as indicated in RC2.4 and RC2.5.

SI file - We have included a new Section S7 where the impact of non-linearities and their connections with dynamical strain are made more explicit.

Reply to Reviewer #3

Original report by Reviewer #3

Review of: Dynamically-enhanced strain in atomically thin resonators

The authors describe an interesting experimental study of the strain in a resonating graphene membrane. For this purpose they employ for the first time a simultaneous measurement of the Raman phonons and the mechanical motion of the membrane. Interestingly they find evidence that the strain in the graphene increases at higher driving forces.

The experimental results are new and relevant and certainly worth publication, however I have some difficulties with the presented analysis of the data that needs to be addressed before publication, since the conclusions might be inaccurate. In particular the conclusion: "These anomalously large strains exceed the time-averaged values predicted for harmonic vibrations with the same root mean square (RMS) amplitude by more than one order of magnitude" needs stronger backup.

Our general reply - We are grateful to the reviewer's for his enthusiastic assessment and recommendation to publish our work. We believe that her/his comments and queries - that partly overlap with those of Reviewer 2 - have helped us improving our manuscript significantly and more explicitly stress its key results and novelty. In the following, we address the reviewer's comments on a point by point basis.

RC3.1a - When a membrane resonates around a flat equilibrium position $z_0=0$, I agree with the authors that the time-averaged strain increases with driving amplitude as indicated by equation (S5), because both for positive and negative deflections the strain increases.

However, when the membrane resonates around a bent driving amplitude, with z_0 not equal to zero (and $z_0 > \sqrt{2} z_{rms}$), $z(t)=z_0+\sqrt{2}z_{rms}\cos(\omega t)$ and it is not so obvious why the average strain in the membrane increases. The membrane will move around a certain equilibrium point in Fig. S6 and its strain will increase half of the time (w.r.t. the average strain), but be lower than the average strain the other half of the time.

AR3.1a - In our SI file, we derive the static strain using equation S3 that assumes a parabolic profile with a central deflection ξ . Now, if the membrane oscillates harmonically relative to this static profile, we simply have to temporally average $\xi^2(t) = (\xi + \sqrt{2}z_{rms}\cos(\Omega t + \varphi))^2$ over one oscillation period to get ϵ_d^h , the dynamical strain estimated in the harmonic regime. In addition to the static strain component ($\propto \xi^2$), there is, as the reviewer points out, a term proportional to $\xi \times z_{rms}\cos(\Omega t + \varphi)$ but this term averages out to zero. Still, there remains a non-vanishing term equal to $2 \times z_{rms}^2 \cos^2(\Omega t + \varphi)$ that yields equation S5 (now listed as equation S18 in our revised SI file). Since $\epsilon_d^h = \frac{2}{3}(z_{rms}/a)^2 = 6 \times 10^{-6}$ (for $z_{rms} = 9$ nm and $a = 3$ μ m is the drum radius), we come to the conclusion that the dynamical strain estimated in the harmonic regime is almost two orders of magnitude smaller than the measured dynamical strain $\epsilon_d \approx 3 \times 10^{-4}$ for $z_{rms} = 9$ nm and $a = 3$ μ m (Fig. 2 and 3). Let us note that even if we were to resolve the time-dependent dynamically-enhanced strain, its maximum (peak) value would at most be $\epsilon_d^{Max} = \pm \frac{4\sqrt{2}\xi z_{rms}}{3a^2} \approx 1 \times 10^{-4}$ (considering that $\xi=40$ nm and 60 nm at $V_{DC} = -6$ V and -8 V, respectively) a value that is close, but still smaller by a factor of 2 to 3 than the time averaged dynamical strains we measure experimentally.

CM3.1a - We have written a new section (S7: Dynamical strain and non-linearities) where section S7a: “Dynamical strain induced by harmonic vibrations” includes Eq. (S18), previously listed as Eq. (S5). The text has been expanded, in particular with the following sentence:

“The time-averaged harmonic dynamical strain ϵ_d^h can be estimated by inserting $\xi(t)$ into Eq. (S3) and averaging over one oscillation period. Since the crossed term $2\sqrt{2}\xi z_{rms} \cos(\Omega t + \varphi)$ averages out to zero, we obtain: $\epsilon_d^h = \frac{2}{3}(z_{rms}/a)^2$ ”

RC3.1b - It is not clear to me, why under these conditions, the dynamic motion will significantly increase average strain. I would only expect a broadening of the Raman peak without a significant shift in the average value. This broadening is also observed and might contain the most important information on the strain fluctuations, but is not analysed in as much detail.

AR3.1b - The reviewer is fully right that temporal averaging may lead to a broadening of the Raman features. This sizeable broadening (by 10 %) is observed and discussed in our manuscript (Fig. 2d). However, as the reviewer also rightly points out, we do essentially observe a downshift of the Raman features. As discussed in details in our reply to Reviewer #2, this downshift is assigned to the complex mode profiles that develop in the membrane as it oscillates non-linearly. Non-linearities alter the membrane profile such that the oscillations can be rectified as suggested by the large discrepancy between ϵ_d^h and the measured ϵ_d and by the fact that ϵ_d is closer to (but still larger than) the peak value $\epsilon_d^{Max} = \frac{4\sqrt{2}\xi z_{rms}}{3a^2}$ introduced in our reply to comment 1. In the case of non-linearly rectified oscillations, one can expect a sizeable time-averaged downshift of the G and 2D-mode features with moderate broadening, as is the case in our experiments.

For the sake of completeness, we do show below the broadening of the 2D-mode feature observed when recording the data shown in Figure 3. A slight 2D-mode broadening by about 1 cm^{-1} (that is close to 5 % of the 2D-mode linewidth) is indeed observed beyond experimental uncertainties ($\pm 0.25 \text{ cm}^{-1}$). In contrast, the associated downshift reaches up to 2.5 cm^{-1} . Similar observations are made in Fig. 2, where the G- and 2D-mode linewidth are already shown.

Figure 2 - 2D-mode linewidth (namely the full width at half maximum of the 2D-feature as explained in supplementary section S2) as a function of the drive frequency $\Omega/2\pi$. This plot is obtained from the Raman data shown in Figure 3 of the main text and now shown ad Fig. S16.

CM3.1b -

Main text - We have added the following sentence in the discussion section : “Finally, the fact that $\Delta\Gamma_{G,2D}$ (Fig. 2 d and S16) is smaller than the associated $\Delta\Omega_{G,2D}$ (Fig. 2c and 3a) suggests that the oscillations of ϵ_d are rectified under strong non-linear driving, an effect that further increases the discrepancy between the time-averaged ϵ_d we measure and ϵ_d^h .”

SI file - The figure shown above has been added to the SI file as Fig. 16.

RC3.2a - Equation (S5) seems only valid for the flat membrane configuration. For a membrane that is bent by electrostatic force induced by V_{dc} , the effect of z_{rms} on strain is roughly indicated by the slope of the curve in Fig. S6, so the induced strain will not only depend on z_{rms} but also on the value of V_{dc} (and z_0).

AR3.2a - This point is partly addressed in our reply to RC3.1a by the reviewer. Eq. (S18) (formerly listed as Eq. (S5)) is in fact valid in the case of a bent membrane.

The reviewer is right that the slope $\frac{\partial \epsilon_s}{\partial \xi} = \frac{4 \xi (V_{dc})}{3 a^2}$ (with ϵ_s , the static strain and ξ , the static displacement) is smaller at $V_{dc} = -6$ V than at $V_{dc} = -8$ V. However, care has to be taken because Fig. S7 (previously listed as Fig. S6 in the revised SI file) is based on a study in the static regime and we are operating in a strongly non-linear regime. We would like to stress, as indicated in our reply to Reviewer #2 that the enhanced dynamical strain (ϵ_d) we measure cannot be understood as resulting from harmonic vibrations (Eq. (S18) and comment RC3.1a by the reviewer and reply AR3.1a) but rather from a complex mode profile that develops in the heavily non-linear regime explored here. Non-linearities lead to large mode profile gradients (as observed for instance in bulkier membranes in (Ref 52, PRL **122**, 154301 (2019) by F. Yang *et al.*), see also our new Supplementary Section S7). These gradients, averaged over the whole membrane, yield enhanced ϵ_d . In other words, while it is true that the slope $\frac{\partial \epsilon_d^h}{\partial z_{rms}} = \frac{4 z_{rms}}{3 a^2}$ as derived from Eq. (S18) is proportional to z_{rms} , we have chiefly used Eq. (S18) to establish that the measured ϵ_d largely exceed ϵ_d^h . We then assign this discrepancy to non-linearities and in particular geometrical non-linearities (See Supplementary Section S7b). We acknowledge that referring extensively to Eq. (S18) may have been

misleading and our intention was certainly not to attempt modelling our results using a harmonic model and Eq. (S18).

CM3.2a - In our revised SI file (Sec. S3), we have explicitly written that $\frac{\partial \epsilon_s}{\partial \xi} = \frac{4}{3} \frac{\xi(V_{dc})}{a^2}$ (now listed as eq. (S5)) and added the following text: “To obtain a larger sensitivity towards strain, dynamical Raman measurements were performed at sufficiently large V_{dc} to yield sizeable ξ , while at the same time maintaining the graphene drum at reasonable distance (>200 nm) from the Si/SiO₂ substrate and avoiding sample collapse and limiting electrostatic non-linearities.”

We have also included a detailed section (S7, “Dynamical strain and non-linearities”) where we discuss the direct relationship between geometrical non-linearities and dynamical strain. This section also provides a rationale for the discrepancy between the dynamical strain expected in the linear regime for harmonic oscillations (ϵ_d^h) and our measurements of anomalously large ϵ_d .

RC3.2b - Like in the previous comment, the broadening of the peak is expected to depend on the product of the slope of the curve in Fig. S6 and z_{rms} .

AR3.2b - We do indeed observe in Fig. 2d that the broadening increases with the driving strength, or equivalently with z_{rms} . Following the reviewer’s comment, and as outlined just above, we observe a larger dynamically-induced broadening at $V_{dc} = -8$ V (see in Fig 2d) than at $V_{dc} = -6$ V (Fig. S16). Here also, care has to be taken because we are operating in a non-linear regime, where a simple reasoning that assume harmonic or quasi harmonic oscillations may be misleading.

CM3.2b - The sentence “This phonon softening is accompanied by spectral broadening by up to 10-15 % (Fig. 2d)” now reads “This phonon softening is accompanied by spectral broadening by up to 10-15 % (Fig. 2d) that increases with z_{rms} ”.

RC3.2c - The use of equation (S5) to estimate the expected strain increase seems not correct and can explain why the authors come to conclusion that “These anomalously large strains exceed the time-averaged values predicted for harmonic vibrations with the same root mean square (RMS) amplitude by more than one order of magnitude”, that might be erroneous.

AR3.2c - We kindly beg to differ with the reviewer on this statement and hope that we have clarified all misunderstandings in our detailed replies (AR1a,b and AR2a,b and changes made presented just above).

RC3.3 - It has been reported that the strain measured by Raman is different from the strain measured from the displacement of the graphene membrane due to hidden area effects in this work: <https://journals.aps.org/prl/abstract/10.1103/PhysRevLett.118.266101>

Since the authors also compare these two methods to analyse strain of graphene, it would be of interest to the reader if the authors can comment whether they see a similar discrepancy between strain estimated from membrane deflection and strain estimated from Raman.

AR3.3 - We thank the reviewer for this interesting comment. We are aware of the results discussed in the PRL paper by Nicholl *et al.* that the reviewer refers to. In this work, due to crumpling effects, the local

static strain probed using Raman spectroscopy is “smaller” than the “geometrical static strain” that is obtained after measuring the membrane deflection interferometrically. In other words, the interferometric measurements probe not only the changes in C-C bond length (as Raman scattering spectroscopy does) but also the stress-induced “flattening” of the initially crumpled graphene, i.e., the hidden area in graphene. In “strain-engineered” samples, designed specifically by the authors to provide a sufficient built-in tension that overcomes initial static crumpling, interferometric and Raman measurements provide identical strain/stress relationships.

In our study, we have deliberately chosen to avoid extensive comments on crumpling and rippling effects. The reason is that these effects are known to affect graphene membranes obtained by chemical vapor deposition (CVD) and processed with resists and solvents, as is the case in the reference cited by the reviewer. In contrast, **our fabrication process is based on dry-transfer of mechanically exfoliated natural graphite flakes and is therefore “resist free” and prone to minimize crumpling and rippling.** Along these lines, our measurements (see section S5 for displacement calibration) confirm that our drums are pristine, i.e., that to experimental accuracy their effective mass and Young’s modulus match the tabulated values for pristine monolayer graphene, such that crumpling and rippling can be neglected. In a previous study (Metten *et al.*, *Phys. Rev. Applied* 2014, cited as Ref. 31), we had investigated similar pristine graphene blisters made from dry transfer and found a good agreement between interferometric and Raman-based methods. Consistently, as discussed in Supplementary Section S5, we find a very good agreement between the static deflection ξ estimated through Raman scattering measurements and through an interferometric measurements (see also our reply to Statement 2 by Reviewer#2), which is another way of saying that to experimental accuracy, we are not probing signatures of crumpling or rippling effects.

CM3.3 -

Methods - We have added the following sentence and cited the 2017 PRL paper by Nicholl *et al.* : “Our dry transfer method minimises rippling and crumpling effects~\cite{Nicholl2017}, resulting in graphene drums with intrinsic mechanical properties (see Ref.~\onlinecite{Metten2014} and Supplementary Section S5 for details)”

Supplementary Section S5 - We have added the following paragraph and cited the PRL paper by Nicholl along with another earlier paper on crumpling and rippling effects (R. Nicholl *et al.*, *Nature Communications* 6, 8789 (2015)). “In addition, graphene drums and blisters, in particular when made from wet-transfer of graphene layers grown by chemical vapor deposition (CVD), are known to exhibit rippling and crumpling~\cite{Nicholl2015}. The resulting hidden area effects lead to discrepancies between the levels of stain determined through Raman and interferometric measurements~\cite{Nicholl2017} and thus affect our displacement and strain calibration. Here, the excellent agreement between calibration methods C_2 and C_3 demonstrates that our graphene drums are immune from hidden area effects, as previously observed in our blister test on pristine suspended graphene, where a Young’s modulus matching that of bulk graphite was found~\cite{Metten2014}”

RC3.4 - The three comments above are the key points. Some more minor comments: RC3.4a. There have been discrepancies in the definition in of the 2D Young’s modulus of graphene. From 3D mechanics it is found that: the ration of the biaxial tension and biaxial strain is given by: $T_{\text{biaxial}}/\epsilon_{\text{biaxial}}= E_{3D} \cdot t / (1-\nu)$. Where E_{3D} is the 3D Young’s modulus, t the membrane thickness and ν the Poisson ratio of an isotropic material. At the start of page 4 of the manuscript it is given that

$T_0/\epsilon_{s0} = E_{1LG}/(1-\nu^2)$. That E_{1LG} doesn't seem to agree with the common definition $E_{2D} = E_{3D}t$, can the authors explain why?

AR3.4a - The reviewer is right and we are grateful that she/he has pointed out this minor error.

CM3.4a - We have corrected the expression that connects the tension to the biaxial strain ($T_0 = \frac{E_{1LG}}{1-\nu} \epsilon_{biaxial}$, with $E_{1LG} = E_{3D}t$) throughout our revised manuscript and SI file, in particular in sections S6 (Mechanical response of driven graphene drums) and S8 (Effects of laser-induced heating, previously S7 in the original submission). Since the Poisson ratio is $\nu \approx 0.16$, this leads to minor corrections in the levels of built-in strain we estimate.

RC3.4b - The authors explain that since $V_{dc}^2 \gg V_{ac}^2$, and because symmetry breaking effects are small, changes in the equilibrium position and nonlinearity at higher V_{ac} are small. However, nonlinear electrostatic force effects can also arise because the condition $z_{rms} \ll d$ is violated ($F_{el} = A \epsilon_0 V^2 / [2(d-z)^2]$). Since gap $d = 250$ nm and initial deflection due to the DC voltage is 60 nm, and z_{rms} is of the order of 10 nm this inequality might not hold in. The authors are encouraged to comment if these electrostatic nonlinearities are relevant.

AR3.4b - We thank the reviewer for this valuable comment and agree with her/him that electrostatic nonlinearities may become important even if, in our case $z_{rms} \ll d$. In our work, non-linearities, and in particular geometrical non-linearities, will favor the observation of dynamically-enhanced strain. As discussed in our reply to reviewer#2, strong non-linearities may be more challenging to observe under photothermal driving.

Let us now focus on electrostatically-induced non linearities. Considering the work by Davidovikj *et al.* (cited as Ref. 7), we can estimate the ratio between the non-linear intrinsic cubic stiffness of graphene ($k_3^0 > 0$) and the non-linearly-induced softening term arising from the variations of the gate capacitance with the distance between the vibrating graphene layer and the Si backgate ($k_{3,soft} < 0$, see Eq. (28) in the SI file of Ref. 7). Using our experimental parameters, we find that the ratio $k_{3,int}/|k_{3,soft}|$ is near 3.1 at $V_{dc} = -6$ V (Fig. 3) and reaches 1.0 at $V_{dc} = -8$ V (Fig. 2), such that non-linear softening should certainly be taken into account. This is evidenced by the fact that we do indeed see non-linear softening for $|V_{DC}| > 6$ V as discussed in Figs 2,3 and in our SI file (Fig. S12 in our revised version). At the same time, we estimate that the gate-induced second order spring constant is large enough and dominates the intrinsic and electrostatically-induced third order terms in Eq. (S15) at large enough V_{dc} , justifying the negative effective third order non-linearity $\tilde{\alpha}_3$ extracted in Fig. S12a,

Let us note that the non-linear terms introduced above are distinct from the third order stiffness term introduced in our new section S7 that is due to geometrical non-linearities (i.e., the displacement-dependent tension of the membrane). In any event, our focus in this work is to demonstrate that Raman scattering spectroscopy can act as a very sensitive probe of dynamical strain fields and that, conversely, electrostatically-induced strain can be used to modulate intrinsic quantum degrees of freedom (here phonons) in graphene. Future studies will focus on the physical origin of the non-linearities, of the amplitude saturation, as well the on the mapping of non-trivial mode profiles.

CM3.4b -

Main text - We are now explicitly referring to electrostatic non-linearities in the discussion section and are citing the following additional reference (Ref. 55) by B. Sajadi *et al.*, Journal of Applied Physics **122**, 234302 (2017)).

SI File - Section S6b: “Non-linear mechanical response” has been expanded with the following text based on our reply above: “Non-linearities can be either be i) intrinsic to graphene, e.g. due to its cubic spring constant~\cite{Lee2008} but also ii) electrostatically-induced by the dependence of the gate capacitance on the distance between the vibrating graphene drum and the Si backgate~\cite{Davidovikj2017} or iii) geometrically induced by the displacement-dependent tension induced by the vibrations of the drum~\cite{Schmid2016}. For instance, using Eq. (12) and (26) in the supplementary information of Ref.~\onlinecite{Davidovikj2017}, we can estimate that the ratio between the third order intrinsic stiffness of graphene and the gate-induced third order softening term is close to 3 at $V_{dc} = -6 V$ and near unity at $V_{dc} = -8 V$. At the same time, we estimate that the gate-induced second order spring constant (α_2) is large enough such that Eq. (S15) yields $\tilde{\alpha}_3 \approx -\frac{10\alpha_2^2}{9\Omega_0^2} \approx -1 \times 10^{32} \sim m^2 s^{-2}$ at $V_{dc} = -8 V$ This value is in good agreement with the experimental value extracted from a fit of the backbone curve in Fig.12a..”

REVIEWER COMMENTS

Reviewer #1 (Remarks to the Author):

The authors have made their efforts to address the concerns and questions raised by the reviewers. Although the quality of the manuscript is increased by the efforts made by the authors, I'm still not fully convinced by the authors.

(1)The Dirac point of the device. In their response, the authors claim that 'the electrostatic force vanishes when the sample is neutral such that we expect a symmetric behavior with respect to the charge neutrality point'. Such statement is incorrect. Even at charge neutrality point, monolayer graphene flake with the size described in the manuscript still conducts, thus the electrostatic force will not vanish.

As I mentioned in the first round review, which the authors also agree, the symmetric gate dependence is due to electrostatic force induced by gate voltage rather than transport properties. So Figure 1c cannot support the conclusion that 'the charge neutrality point is, in our work at 0.75 V'.

(2)In the method part, the authors added 'In the dynamical regime, the RMS modulation of the doping levels induced by the application of V_{ac} is typically two orders of magnitude smaller than the static doping level and can safely be neglected'. I'm wondering whether the authors considered the change of the doping level contributed by the change of the gate capacitance, since applying V_{ac} will shift the equilibrium position (as shown in Fig. 2e).

(3)One minor suggestion. Since the graphene drum resonator is strongly driven into nonlinear regime, nonlinear dissipation, which is commonly observed in nanomechanical resonators, should play a significant role. Could the authors briefly comment on the influence of the nonlinear damping?

I would like to ask the authors to provide more detailed discussions to address the concerns above before I can recommend its publication in Nature Communications.

Reviewer #2 (Remarks to the Author):

The manuscript reports nano electro mechanical studies of graphene drums using Raman spectroscopy. The main claim of the manuscript is a very large value of dynamical strain. This is the only novel claim that remains unsubstantiated in terms of the underlying physics.

I had asked authors several questions in the first round of review to urge them to provide a more detailed understanding of microscopic mechanism. My questions included estimates about electrostatic non-linearities and possibility of putting a bound using estimates of thermo mechanical actuation. I did not ask them to do additional experiments. They have chosen to deflect these questions as not relevant to present experiment.

The authors observe a large dynamical strain but just attribute everything to non-linearities broadly. They even invoke intrinsic localised modes. As this sub field is mature a new claim needs a fuller description for it to be publishable in a journal like Nature Communications. Even after revision their manuscript does not rise to the threshold of publication in Nature Communications. It should not be published in Nature Communications.

Reviewer #3 (Remarks to the Author):

The authors have extensively addressed my comments, and provided satisfactory changes and improvements to the manuscript. The discussed experimental technique is useful and novel. It can be applied to a large number of 2D materials and by itself justifies publication of this work. In addition the inferred dynamically enhanced strain in graphene is interesting, and will stimulate

follow-up investigations to unveil its microscopic origin.

In my view, this thorough work is certainly of interest to the readers of Nature Communications and I therefore recommend its acceptance for publication.

Peter Steeneken

In the following, we reply to the reviewers and address their comments on a point by point basis.

Reviewer #1 (Remarks to the Author):

The authors have made their efforts to address the concerns and questions raised by the reviewers. Although the quality of the manuscript is increased by the efforts made by the authors, I'm still not fully convinced by the authors.

Our reply: We are grateful to Reviewer 1 for her/his positive assessment and interesting questions. In the following we address her/his queries in full.

Comment (1)

The Dirac point of the device. In their response, the authors claim that 'the electrostatic force vanishes when the sample is neutral such that we expect a symmetric behavior with respect to the charge neutrality point'. Such statement is incorrect. Even at charge neutrality point, monolayer graphene flake with the size described in the manuscript still conducts, thus the electrostatic force will not vanish. As I mentioned in the first round review, which the authors also agree, the symmetric gate dependence is due to electrostatic force induced by gate voltage rather than transport properties. So Figure 1c cannot support the conclusion that 'the charge neutrality point is, in our work at 0.75 V'.

Our reply: This question by the reviewer in fact relates to whether or not one can measure the unintentional doping level of graphene using the gate-tunable mechanical resonance frequency.

We agree with the reviewer that electron transport measurements of the gate-dependent longitudinal conductivity as a function of the source and drain bias are commonly used to determine the charge neutrality point (CNP) of graphene and its unintentional doping level. With our experimental geometry, electron transport measurements of the CNP cannot be performed since the both supported and suspended regions of our graphene monolayers are electrically contacted by the metal pads.

Let us stress that all our measurements are performed on pristine suspended graphene, which is well known to have a vanishingly small unintentional charge carrier density ($< 10^{11} \text{ cm}^{-2}$), importantly also with minimal charge inhomogeneity (the well-known "electron and hole puddles" largely discussed in the literature). The quasi-neutrality of such samples is assessed using spatially-resolved Raman measurements.

With such a quasi-undoped sample, we expect, as shown in Figure 1c, a symmetric evolution of the mechanical resonance as a function of V_{dc} with respect to $V_{\text{dc}} \approx 0$, where graphene is neutral and experiences minimal tensile stress. At $V_{\text{dc}} = 0$, quasi-neutral graphene only undergoes built in tension T_0 chiefly due to the native strain in the membrane. In addition, T_0 might in part be due to a residual electrostatic force arising from the interaction between the minute amount of residual dopants (possibly with a spatially averaged contribution from e-h puddles) on graphene and some trapped charges that may be present at the Si back gate surface.

In our experiments the symmetry point in Fig. 1c is observed at $V_{\text{dc}} = 0.75 \text{ V}$, i.e., very near $V_{\text{dc}} \approx 0$, as in previous similar studies on quasi-neutral graphene (for instance in Ref. 4 and Ref. 5, Fig. 3). As

shown in Ref. 4 (Fig. 5c), when graphene is nearly pristine (that is, it is devoid from contamination residues and unintentional doping), the minimal conductivity point observed through transport measurements at V_{dc}^D (D referring to the Dirac point in graphene) and the gate bias V_{dc}^0 relative to which the gate-dependent mechanical frequency is symmetric are very close. For these reasons, we estimated the unintentional doping level through our measurement of V_{dc}^0 .

Here, in keeping with the reviewer comment we would like to stress that the correspondence between V_{dc}^0 and V_{dc}^D is generally not straightforward in “non-pristine” samples. For instance, when a suspended graphene layer is unintentionally doped and/or contaminated with fabrication residues (as in Ref. 4, Fig. 4 and 5a), combined transport and mechanical measurements yield significant discrepancies between V_{dc}^0 and V_{dc}^D .

Changes made: Since the exact determination of the charge neutrality point is not essential in our study and may in fact distract the reader, we have done the following changes in our revised manuscript.

The sentence “*The mechanical resonance frequency (...) displays a symmetric, “U-shaped” behavior with respect to $V_{dc}^0 = 0.75 V$, where graphene is charge neutral (...)*” now writes “*The mechanical resonance frequency (...) displays a symmetric, “U-shaped” behavior with respect to a near-zero DC bias $V_{dc}^0 = 0.75 V$, at which graphene only undergoes a built-in tension.*”

The sentence “*From the value of V_{dc}^0 , we estimate a minute unintentional hole doping below $2 \times 10^{10} \text{ cm}^{-2}$ that is consistent with the intrinsic character of suspended graphene~\cite{Berciaud2009,Berciaud2013}*” has been removed and we have added the following sentence in the methods sections to highlight the quasi neutral character of our suspended graphene samples : “*Pristine suspended graphene, as used here, is well-known to have minimal unintentional doping ($< 10^{11} \text{ cm}^{-2}$) and charge inhomogeneity~\cite{Berciaud2009,Berciaud2013}.*”

Comment (2)

In the method part, the authors added ‘In the dynamical regime, the RMS modulation of the doping levels induced by the application of Vac is typically two orders of magnitude smaller than the static doping level and can safely be neglected’. I’m wondering whether the authors considered the change of the doping level contributed by the change of the gate capacitance, since applying Vac will shift the equilibrium position (as shown in Fig. 2e).

Our reply: The reviewer is right that the gate capacitance will be affected by the displacement (be it static or dynamic) of the graphene drum and also by the equilibrium position change. This being said, in our first reply and in our revised manuscript, we have demonstrated that the maximal charge densities applied at the largest DC gate biases $V_{dc} = -10 V$ are near $3 \times 10^{11} \text{ cm}^{-2}$ not sufficient to induce significant fingerprints of doping on the Raman G- and 2d-mode features. In fig. 2e and 3a-iv, we reveal equilibrium position *upshifts* of at most 12 nm, i.e. $\sim 5\%$ of the graphene-gate distance. These upshifts of the membrane marginally decrease the gate capacitance of the device and further reduce the doping induced by the application of the AC and DC gate biases. As a result, equilibrium position shifts do not induce measurable fingerprints of reduced doping on graphene.

Changes made: we have added the sentence “*Similarly, the reduction of the gate capacitance induced by equilibrium position upshifts discussed in Fig. 2e and Fig. 3a-iv do not induce measurable fingerprints of reduced doping on graphene.*” in the methods section.

Comment (3)

One minor suggestion. Since the graphene drum resonator is strongly driven into nonlinear regime, nonlinear dissipation, which is commonly observed in nanomechanical resonators, should play a significant role. Could the authors briefly comment on the influence of the nonlinear damping?

Our reply: Indeed, non-linearities also include non-linear damping, i.e., a damping rate that depends on the displacement of the drum. For the sake of simplicity, we had chosen not to mention non-linear damping in our discussion. The reason for this is that we did not intensively focus on the damping in our drums but rather on the maximum RMS displacements (and dynamically-enhanced strain) we could achieve.

Changes made: In supplementary section S6, we consider the Reviewer's suggestion with the following footnote, listed as Ref. 27 in the SI file. *"In principle, Eq. (S9) could include other non-linear contributions, and in particular a non-linear damping term (proportional to $\dot{z}z^2$)\cite{Eichler2011a,Imboden2013}. Non-linear damping may broaden the frequency-dependent mechanical susceptibility of our drums, reduce its resonant amplitude and may thus act against the enhancement of ϵ_d . As a result, more pronounced dynamically-induced strain enhancement could be achieved provided non-linear damping is minimized."* We also cite two references, A. Eichler et al., Nat. Nanotechnol. **6**, 339 (2011) and M. Imboden et al., Appl. Phys Lett **102**,103502 (2013), listed as Refs. 38 and 39 in our revised SI file, respectively.

Reviewer #2 (Remarks to the Author):

The manuscript reports nano electro mechanical studies of graphene drums using Raman spectroscopy. The main claim of the manuscript is a very large value of dynamical strain. This is the only novel claim that remains unsubstantiated in terms of the underlying physics.

I had asked authors several questions in the first round of review to urge them to provide a more detailed understanding of microscopic mechanism. My questions included estimates about electrostatic non-linearities and possibility of putting a bound using estimates of thermo mechanical actuation. I did not ask them to do additional experiments. They have chosen to deflect these questions as not relevant to present experiment.

The authors observe a large dynamical strain but just attribute everything to non-linearities broadly. They even invoke intrinsic localised modes. As this sub field is mature a new claim needs a fuller description for it to be publishable in a journal like Nature Communications. Even after revision their manuscript does not rise to the threshold of publication in Nature Communications. It should not be published in Nature Communications.

Our reply:

We thank the reviewer for evaluating our revised manuscript. Reviewer 2 validates our experimental methodology, allowing to quantitatively determine dynamical strain in graphene and, as a second step to unravel that dynamical strain is anomalously high under strong non-linear driving. Her/his remaining comment is on the microscopic origin of this enhancement.

We have appreciated the constructive criticism by Reviewer 2 in her/his first report and strived to address all her/his comments and queries in full (8 in total + two statements on which we also commented) in our thoroughly revised manuscript. Several points raised by Reviewer 2 in her/his first report overlap with the report by Reviewer 3. Unfortunately, while Reviewer 3 clearly hailed our efforts to improve and clarify our paper (see below), Reviewer 2 does not mention that we have sequentially addressed all the numerous points that she/he raised, and included quantitative discussions in our manuscript and SI file.

Therefore, we have to say that we are somewhat bemused by the second report of reviewer 2, which we believe is a bit of a judgement call. Hereafter, and we would kindly like to rebut his criticism.

1) On photothermal vs electrostatic actuation

We have addressed this point (and the related comments raised by reviewer 3) in details in our reply to the first round of reports and in our revised manuscript.

First, in our manuscript and SI file, we have added discussions to show that electrostatic actuation could induce sizeable non-linearities (see, in particular supporting Section S6b, page 23) and we have quantified these non-linearities (as also suggested by Reviewer 3; see also point 2)).

Second, we have cited relevant papers (Ref. 10, 11 and 39 in our manuscript) to mention the shortcomings of photothermal actuation when it comes to achieving large resonant RMS displacements and probe mechanical non-linearities. We can even go further and refer to Fig. 3f in Ref. 8 by J. Lee *et al.*, This figure shows that photothermal actuation leads to significantly more damping

than electrostatic actuation and hence lower resonant RMS amplitudes, which is another reason to favor electrostatic actuation.

Change made: the sentence “We have chosen electrostatic rather than photothermal actuation~\cite{Sampathkumar2006} to attain large RMS amplitudes while at the same time avoiding heating and photothermal backaction effects~\cite{Barton2012,Morell2019}, self-oscillations, mechanical instabilities and sample damage” now writes “We have chosen electrostatic rather than photothermal actuation~\cite{Sampathkumar2006} to attain large RMS amplitudes while at the same time avoiding heating and photothermal backaction effects~\cite{Barton2012,Morell2019}, possibly leading to additional damping~\cite{Lee2018}, self-oscillations~\cite{Barton2012}, mechanical instabilities and sample damage”.

For the sake of completeness, in our reply to the first round of reports, we had also indicated that the “shortcomings” of photothermal actuation (in particular photothermal backaction) may become relevant to investigate physical phenomena (thermal transport, for instance) outside the scope of our paper.

All in all, photothermal driving would not allow us to drive our drums non-linearly in a sufficiently strong and controlled fashion.

We therefore refute the following comment: “My questions included estimates about electrostatic non-linearities and possibility of putting a bound using estimates of thermo mechanical actuation. [...] They have chosen to deflect these questions as not relevant to present experiment.”

2) On the microscopic origin of dynamically-enhanced strain

We do not conceal that a more quantitative microscopic description of dynamically-enhanced strain is still desirable. However, we are convinced that this research goes well-beyond the scope of the present manuscript as it necessitates a dedicated experimental study on drums with significantly larger diameter (well-above the optical diffraction limit) to allow probing mechanical mode profiles in greater details. Such samples are technically challenging to fabricate. The Reviewer her/himself stresses that she/he did not request additional experiments.

By introducing an original methodology combining quantitative optical spectroscopy and nano-mechanics, our manuscript reports the first observation of dynamically-enhanced strain under non-linear driving. This new finding is unambiguous and backed-up by a comprehensive set of experiments. We propose a compelling microscopic interpretation, where dynamically-enhanced strain arises from geometrical non-linearities and the emergence of non-trivial, “localised” modes. Contrary to what Reviewer 2 claims, the study of such “localised modes” is not at all a mature field since the first paper on the topic only appeared last year (by F. Yang *et al.* PRL 122, 153301 (2019) on bulkier SiN membranes, listed as Ref. 53). To our knowledge, there is so far no publication on related phenomena in the strictly 2D limit and our paper is an important step towards a better understanding of the strong mechanical non-linearities in 2D materials.

Reviewer 2 asked us to provide a rationale for the proportionality between the non-linear mechanical frequency shift δ and the dynamically enhanced strain ϵ_d (Fig. 2f and 3d). Motivated by this query, we have written section S7 in our revised SI file and summarized this discussion on page 7 (1st column of the revised MS we submitted in July). These additions focus on the specific role of geometrical non-linearities in our experiments. Geometrical non-linearities are distinct from the other sources of non-

linearity (intrinsic, electrostatic,...) that are also discussed in Section S6 and in our MS. Unfortunately, the reviewer does not comment on these major revisions made in our MS and SI files.

Graphene electro-mechanical resonators are highly non-linear systems and it is notoriously challenging to disentangle contributions from various sources of non-linearity. We believe that identifying the main non-linearities (intrinsic, electrostatic, geometrical) and estimating their contributions (which we did) is fair. Going beyond this is a daunting perspective and, in the absence of high-resolution mode mapping, is very speculative.

For these reasons we would like to refute this second comment by the reviewer: *“The authors observe a large dynamical strain but just attribute everything to non-linearities broadly. They even invoke intrinsic localised modes. As this sub field is mature a new claim needs a fuller description for it to be publishable in a journal like Nature Communications.”*

Reviewer #3 (Remarks to the Author):

The authors have extensively addressed my comments, and provided satisfactory changes and improvements to the manuscript. The discussed experimental technique is useful and novel. It can be applied to a large number of 2D materials and by itself justifies publication of this work. In addition the inferred dynamically enhanced strain in graphene is interesting, and will stimulate follow-up investigations to unveil its microscopic origin.

In my view, this thorough work is certainly of interest to the readers of Nature Communications and I therefore recommend its acceptance for publication.

Our reply: we sincerely appreciate this enthusiastic report by Reviewer 3, and thank him for evaluating our original and submitted manuscripts.

REVIEWERS' COMMENTS

Reviewer #1 (Remarks to the Author):

The authors have addressed my concerns properly. Thus I recommend its publication in Nature Communications.

Reviewer #2 (Remarks to the Author):

I have read the response of the authors and studied the manuscript. However, my assessment is that the manuscript does not rise to the level I expect from a manuscript in Nature communications still holds. It should not be published in Nature Communications.

The authors claim an experimental observation of large dynamically enhanced strain in a field of graphene nanomechanics that is mature. However, they do not provide a clear physical picture for how non-linearities, of any kind, could lead to the "surprising" observations.

REVIEWERS' COMMENTS

Reviewer #1 (Remarks to the Author):

The authors have addressed my concerns properly. Thus I recommend its publication in Nature Communications.

Author's Reply: We thank reviewer #1 for his/her positive assessment and we are glad that the points he/she raised are now clarified.

Reviewer #2 (Remarks to the Author):

I have read the response of the authors and studied the manuscript. However, my assessment is that the manuscript does not rise to the level I expect from a manuscript in Nature communications still holds. It should not be published in Nature Communications.

The authors claim an experimental observation of large dynamically enhanced strain in a field of graphene nanomechanics that is mature. However, they do not provide a clear physical picture for how non-linearities, of any kind, could lead to the "surprising" observations.

Author's Reply: We thank Reviewer #2 for considering again our manuscript after a second round of revisions. We have strived to answer all the reviewer's queries as thoroughly as possible. We are convinced that the first report by Reviewer #2 helped us clarify our claims and improve our manuscript. We would like to reiterate that unveiling the microscopic details of dynamical strain enhancement is presently without experimental reach and would require another in-depth study on specifically-designed samples.